# Query Routing over Multimodal Knowledge Bases for Retrieval-Augmented Reasoning

## Abstract

Multimodal Retrieval-Augmented Generation (MRAG) has shown promise in mitigating hallucinations in Multimodal Large Language Models (MLLMs) by incorporating external knowledge during generation. Existing MRAG methods typically adopt a static retrieval pipeline that fetches relevant information from predefined Knowledge Bases (KBs), followed by a refinement step. However, these approaches overlook the reasoning and planning capabilities of MLLMs to dynamically determine how to interact with different KBs during the reasoning process. To address this limitation, we propose R1-Router, a novel MRAG framework that learns to decide **when** and **where** to retrieve knowledge based on the evolving reasoning state. Specifically, R1-Router can generate follow-up queries according to the current reasoning step, routing these intermediate queries to the most suitable KB, and integrating external knowledge into a coherent reasoning trajectory to answer the original query. Furthermore, we introduce Stepwise Group Relative Policy Optimization (Step-GRPO), a tailored reinforcement learning algorithm that assigns step-specific rewards to optimize the reasoning behavior of MLLMs. Experimental results on diverse open-domain QA benchmarks spanning multiple modalities demonstrate the strong effectiveness of R1-Router, achieving improvements of more than 7% over competitive baselines. Further analysis reveals that R1-Router adaptively and effectively routes queries to the appropriate modalities during inference, while leveraging knowledge from multimodal KBs to reduce redundant retrieval steps and enhance both efficiency and accuracy.

## 1 Introduction

Retrieval-Augmented Generation (RAG) methods (Asai et al., 2023; Lewis et al., 2020; Shi et al., 2023) empower Large Language Models (LLMs) to access external knowledge, which functions as a form of "memory" that the models can consult during inference (Bai et al., 2024; Chen et al., 2024c; Zhao et al., 2023), helping to mitigate hallucinations and improve the response accuracy (Chen et al., 2024b; Gao et al., 2023; Li et al., 2022). In real-world scenarios, such an external "memory" often derives from multimodal Knowledge Bases (KBs), such as image-caption pairs, textual documents, and tabular data. These KBs play distinct roles in supporting models in acquiring diverse knowledge for answering queries. Existing Multimodal RAG (MRAG) approaches (Abootorabi et al., 2025; Xia et al., 2024; Zhang et al., 2024) typically perform one-pass retrieval from predefined knowledge bases and aim to refine the retrieved knowledge for enhancing the generation process (Lin et al., 2023). This design limits the ability of Multimodal LLMs (MLLMs) to interact dynamically with different KBs and fails to fully meet the information needs of MLLMs during inference (Shao et al., 2023; Su et al., 2024; Yu et al., 2025; Jiang et al., 2023).

To address this limitation, existing research has explored iterative retrieval strategies to elicit more relevant information that satisfies the knowledge requirements of MLLMs for answering (Nan et al., 2024; Shao et al., 2023; Trivedi et al., 2022). These methods typically prompt MLLMs to decompose the input query, retrieve external knowledge for each sub-query, and incorporate the retrieved evidence into the input context for final answer generation. While effective, these approaches often rely on predefined retrieval pipelines prioritizing a single dominant modality, thus limiting their flexibility in acquiring information from diverse KBs (Li et al., 2024b). To mitigate this rigidity, recent work has proposed modular frameworks that introduce a planner to dynamically adjust retrieval actions based on the current sub-query and route queries to specific KBs (Yeo et al., 2025; Yu et al., 2025).

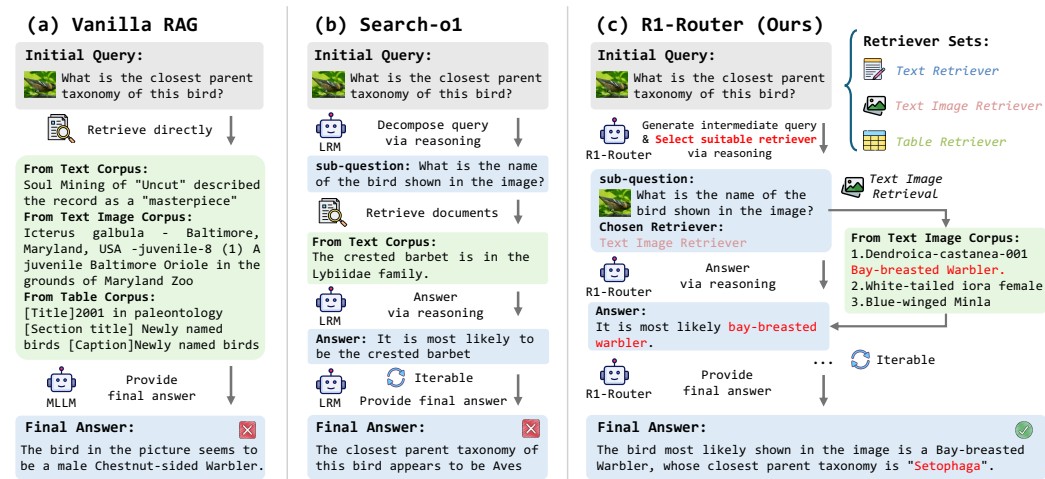

Figure 1: Comparison of Different RAG Architectures.

However, these frameworks primarily depend on the query routing capabilities of MLLMs, without fully exploiting their reasoning abilities (Wu et al., 2024). This reliance constrains the dynamic and adaptive potential of MLLMs in querying different KBs as information needs evolve during the solution of complex RAG tasks (Chen et al., 2023a; Guan et al., 2025).

Furthermore, recent studies have investigated employing Large Reasoning Models (LRMs) (Guo et al., 2025; Li et al., 2025b; Team, 2025; Xu et al., 2025) as backbone models to strengthen the planning and reasoning abilities of the RAG framework (Chen et al., 2025b; Li et al., 2025a). These methods enable LRMs to interact with retrievers during inference, allowing dynamic access to external knowledge when facing uncertain or incomplete information (Li et al., 2025a). In parallel, Reinforcement Learning (RL) techniques have been introduced to train LLMs to decide when to invoke retrieval actions within the reasoning process (Chen et al., 2025a; Song et al., 2025; Zheng et al., 2025; Jin et al., 2025). Despite these advances, current LRM-based approaches remain largely limited to the textual modality and lack multimodal perception capabilities (Qu et al., 2025; Xu et al., 2025), which hinders their ability to handle multimodal queries and to dynamically identify the most suitable knowledge sources for retrieval.

In this paper, we introduce R1-Router, a novel framework that empowers MLLMs to dynamically decide *when* and *where* to acquire external knowledge, by adaptively selecting the next action–query decomposition, modality routing, or answering–based on the current reasoning state. As illustrated in Figure 1, R1-Router allows MLLMs to generate a query when additional knowledge is required, route it to the appropriate knowledge base, and continue reasoning based on previously reasoning results. This iterative process continues until the maximum number of reasoning steps is reached or when the model determines that sufficient information has been gathered. To effectively train R1-Router, we introduce Stepwise Group Relative Policy Optimization (Step-GRPO), which provides step-specific rewards to guide optimization throughout the reasoning trajectory. Experiments across diverse QA tasks validate the effectiveness of R1-Router, yielding substantial gains over existing RAG models. With Step-GRPO, R1-Router performs deeper step-by-step reasoning and generalizes well across various multimodal QA scenarios. Furthermore, R1-Router demonstrates strong flexibility in leveraging multimodal KBs, substantially reducing retrieval steps while maintaining high accuracy in both Visual QA and Table QA tasks.

## 2 RELATED WORK

Multimodal RAG (MRAG) methods (Mei et al., 2025; Sharifymoghaddam et al., 2024; Zhao et al., 2023) extend traditional RAG approaches (Lewis et al., 2020; Shi et al., 2023; Ram et al., 2023) by incorporating multiple Knowledge Bases (KBs) of different modalities, thereby enhancing their capacity to handle more complex and diverse real-world applications (Abootorabi et al., 2025; Caffagni et al., 2024; Yu et al., 2024; Lahiri & Hu, 2024). Typically, these methods route queries to a

predefined KB—such as structured tables or image captions—and perform a one-shot retrieval to provide supporting knowledge for MLLMs during response generation (Chen et al., 2022; Wu et al., 2025b). However, such fixed and rigid retrieval strategies limit MRAG systems in addressing queries that require evidence from multiple knowledge sources, which is critical for producing accurate and reliable answers (Yeo et al., 2025; Liu et al., 2025; Zhao et al., 2023).

To address this challenge, recent studies investigate training or prompting Multimodal LLMs (MLLMs) as agents that adaptively plan retrieval strategies, dynamically selecting the appropriate modality from KBs to support accurate response generation (Yu et al., 2025; Li et al., 2024b; Yeo et al., 2025). For instance, UniRAG (Yeo et al., 2025) introduces a unified retriever–controller architecture that routes queries to modality-specific retrievers based on query type and context. OmniSearch (Li et al., 2024b) employs a modular pipeline that decomposes complex queries into sub-queries and assigns them to modality-appropriate retrievers, thereby integrating heterogeneous knowledge sources. CogPlanner (Yu et al., 2025) leverages LLM-based agents for reasoning-driven query decomposition and retrieval planning, enabling iterative RAG workflows. Nevertheless, these approaches largely rely on prompting or supervised fine-tuning for query routing, instead of exploiting intrinsic reasoning capabilities of MLLMs, which constrains their generalization ability (Chu et al., 2025; Yang et al., 2025).

To further integrate retrieval with reasoning, recent work has applied Reinforcement Learning (RL) (Kaelbling et al., 1996) to optimize text-based RAG models, with methods such as DPO (Rafailov et al., 2023) and GRPO (Shao et al., 2024). DeepRAG (Guan et al., 2025) uses a binary decision tree combined with DPO to train models in deciding whether to respond directly or retrieve first, guided by user preference signals. ReSearch (Chen et al., 2025a) unifies retrieval and reasoning within the GRPO framework, employing final-answer accuracy as feedback to improve performance on complex queries. R1-Searcher (Song et al., 2025) adopts a two-stage, outcome-driven RL training strategy to enhance the search capabilities of LLMs. However, these outcome-based optimization methods depend on coarse-grained training signals and lack fine-grained supervision of intermediate reasoning steps. As a result, they may introduce unnecessary or suboptimal retrieval actions, reducing both the efficiency and stability of the training process.

## 3 METHODOLOGY

This section introduces our proposed method, R1-Router, which enables MLLMs to retrieve knowledge from multiple knowledge sources during reasoning autonomously. We first present an overview of how R1-Router adaptively decides when and where to retrieve relevant information in the reasoning process (Sec. 3.1). We then describe how R1-Router optimizes MLLMs for searching knowledge via the Stepwise Group Relative Policy Optimization (Step-GRPO) algorithm (Sec. 3.2).

### 3.1 THE OVERVIEW FRAMEWORK OF R1-ROUTER

Given an initial query $q_0$ and a set of multimodal knowledge bases (KBs) $D$, R1-Router performs a step-by-step reasoning $\mathcal{R}$ to retrieve relevant information and generate the final answer $a$:

$$(q_0, D) \xrightarrow{\text{R1-Router}(\mathcal{R})} a, \tag{1}$$

where the input query $q_0$ requires knowledge from multiple modalities, such as text and images, to answer, the knowledge bases $D$ provide heterogeneous sources, including collections of image–caption pairs, textual documents, and tabular data. This diversity enables MLLMs to integrate information from different KBs to support accurate query answering. The reasoning process $\mathcal{R}$ executed by R1-Router is organized into $n + 1$ steps:

$$\mathcal{R} = \underbrace{\{t_1, (q_1, s_1), a_1\}}_{\mathcal{R}_1}, \ldots, \underbrace{\{t_n, (q_n, s_n), a_n\}}_{\mathcal{R}_n}, \underbrace{\{t_{n+1}, q_0, a_{n+1}\}}_{\mathcal{R}_{n+1}}. \tag{2}$$

The first $n$ steps, denoted as $\mathcal{R}_{1:n}$, are dedicated to retrieving relevant evidence from different knowledge bases. In the final step, $\mathcal{R}_{n+1}$, R1-Router integrates the accumulated knowledge $a_1, \ldots, a_n$ from previous reasoning steps with the initial query $q_0$ to generate the final answer $a_{n+1}$. Then we provide a detailed description of the knowledge accumulation process $\mathcal{R}_{1:n}$ and the final answer generation step $\mathcal{R}_{n+1}$ of R1-Router.

**Stepwise Knowledge Gathering.** At each reasoning step $\mathcal{R}_i$ $(1 \leq i \leq n)$, R1-Router first generates an intermediate reasoning output $t_i$ to evaluate whether the information accumulated thus far is sufficient to answer the initial query $q_0$:

$$t_i = \text{MLLM}\left(q_0, \mathcal{R}_{1:i-1}\right). \tag{3}$$

Based on the original query $q_0$, the reasoning history $\mathcal{R}_{1:i-1}$, and the current reasoning output $t_i$, R1-Router generates a follow-up query $q_i$ and selects a retriever identifier $s_i$:

$$(q_i, s_i) = \text{MLLM}\left(q_0, \mathcal{R}_{1:i-1}, t_i\right), \tag{4}$$

where $s_i$ is the retriever identifier to determine which KB to search from. The retriever identifier $s_i$ can be **Text Retriever**, **Text Image Retriever**, or **Table Retriever**, each corresponding to a specific retriever that queries a specific knowledge bases from $D$. Using the selected retriever $s_i$, R1-Router queries the corresponding sub-KB $D(s_i)$ with the intermediate query $q_i$ to retrieve relevant evidence $d_i$:

$$d_i = \text{Search}(q_i, D(s_i)). \tag{5}$$

Subsequently, the MLLM then produces a response $a_i$ to answer the intermediate query $q_i$ based on the current context tuple $(t_i, q_i, d_i)$:

$$a_i = \text{MLLM}\left(t_i, q_i, d_i\right). \tag{6}$$

**Final Answer Generation.** Once the the first $n$ step reasoning $\mathcal{R}_{1:n}$ is completed, R1-Router transitions to the final answer generation stage. At the $n + 1$-th step, the MLLM determines that no additional evidence is required and sets the follow-up query to "None" or reaches the maximum reasoning depth. Thus, we replace the "None" with the initial query $q_0$ to get the final answer:

$$\mathcal{R}_{n+1} = \{t_{n+1}, \text{None} \rightarrow q_0, a_{n+1}\}. \tag{7}$$

In this final step, the MLLM generates the answer $a_{n+1}$ according to the integration of the initial query $q_0$, the full reasoning trajectory $\mathcal{R}_{1:n}$, and the reasoning result $t_{n+1}$:

$$a_{n+1} = \text{MLLM}\left(q_0, \mathcal{R}_{1:n}, t_{n+1}\right). \tag{8}$$

## 3.2 Optimizing MLLMs for Query Routing through Step-GRPO

R1-Router first collects a ground truth reasoning trajectory $\mathcal{R}^* = \mathcal{R}_1^*, \ldots, \mathcal{R}_n^*, \mathcal{R}_{n+1}^*$ that can help to produce the correct answer for the initial query $q_0$. It then introduces Step-GRPO, a method that extends Group Relative Policy Optimization (GRPO) (Shao et al., 2024) to optimize MLLMs for retrieving information across different KBs $D$ and performing stepwise reasoning. Specifically, at each reasoning step $\mathcal{R}_i$, the model is conditioned on the previous ground truth reasoning steps $\mathcal{R}_{1:i-1}^*$ and trained to generate the next step reasoning result $\mathcal{R}_i$.

**Stepwise Reward Modeling in R1-Router.** To optimize each reasoning step $\mathcal{R}_i$, R1-Router defines two types of rewards, $r^{(1)}$ and $r^{(2)}$, to guide the model in (1) acquiring appropriate information $d_i$ and (2) producing a more accurate answer $a_i$ within the exploration trajectory.

First, $r^{(1)}$ encourages the model to ask more targeted queries and correctly route them to relevant KBs during the reasoning step $\mathcal{R}_i$ $(1 \leq i \leq n)$. This involves two components: (i) a query reward $r_{\text{ask}}(q_i)$, which measures the semantic similarity between the generated query $q_i$ and the corresponding golden query in $\mathcal{R}_i^*$ using the BGE-M3 embedding model (Chen et al., 2024a); and (ii) a routing reward $r_{\text{route}}(s_i)$, which evaluates whether the model correctly selects an appropriate retrieval to search from a relevant subset of the KB. The overall reward $r^{(1)}$ is defined as:

$$r^{(1)} = r_{\text{format}}(q_i, s_i) \times (\alpha r_{\text{ask}}(q_i) + \beta r_{\text{route}}(s_i)), \tag{9}$$

where $\alpha$ and $\beta$ are hyperparameters balancing the importance of query relevance and correct routing. The formatting reward $r_{\text{format}}(q_i, s_i)$ ensures that both the query and the retriever identifier are enclosed in special tokens, respectively.

Second, to optimize the answer $a_i$ at each step $\mathcal{R}_i$, we define the following reward:

$$r^{(2)} = r_{\text{format}}(a_i) \times r_{\text{answer}}(a_i), \tag{10}$$

where $r_{\text{answer}}(a_i)$ evaluates whether the generated answer $a_i$ is correct. We assess the answer quality by using accuracy and F1-Recall (Li et al., 2024b). The F1-Recall is applied to intermediate answers

$a_i$ ($1 \leq i \leq n$), which often correspond to long and complex LLM-generated references. And, for the final answer $a_{n+1}$, we use the accuracy for evaluation, due to the short golden reference. The formatting reward $r_{\text{format}}(a_i)$ enforces that the answer is enclosed in the special tokens. More details on the fine-grained reward design are provided in Appendix A.4.

**Step-GRPO Objective.** To optimize the policy model for multi-step reasoning tasks effectively, R1-Router adopts a Step-GRPO objective that explicitly computes the policy advantage at each intermediate reasoning step. Given a query $q_0$ that requires $n$ reasoning steps $\mathcal{R}$, Step-GRPO samples a set of outputs at each intermediate step $i$ and minimizes the following objective:

$$\mathcal{L} = \sum_{i=1}^{n} [\mathcal{L}_{\text{GRPO}}((q_0, \mathcal{R}^*_{1:i-1}), r^{(1)}) + \mathcal{L}_{\text{GRPO}}((q_i^*, d_i^*), r^{(2)})] + \mathcal{L}_{\text{GRPO}}((q_0, \mathcal{R}^*_{1:n}), r^{(2)}), \quad (11)$$

where $q_i^*$ and $d_i^*$ denote the golden query and retrieved documents from the $i$-th golden reasoning step $\mathcal{R}_i^*$. The final term, $\mathcal{L}_{\text{GRPO}}((q_0, \mathcal{R}^*_{1:n}), r^{(2)})$ corresponds to the $(n+1)$-th step that focuses solely on generating the final answer $a$ based on the initial query $q_0$ and the full reasoning trajectory $\mathcal{R}_{1:n}$. Each GRPO loss term $\mathcal{L}_{\text{GRPO}}(x, r)$ is computed over a batch of sampled trajectories from the old policy model $\pi_{\theta_{\text{old}}}$, given an input $x$ and reward $r$:

$$\begin{aligned}
\mathcal{L}_{\text{GRPO}}(x, r) = -\frac{1}{G} \sum_{k=1}^{G} \frac{1}{|\mathcal{O}_k|} \sum_{t=1}^{|\mathcal{O}_k|} \Bigg[ \min \Bigg( &\frac{\pi_\theta(o_{k,t} \mid x, o_{k,<t})}{\pi_{\theta_{\text{old}}}(o_{k,t} \mid x, o_{k,<t})} \hat{A}_{k,t}(r), \\
&\text{clip}\Big(\frac{\pi_\theta(o_{k,t} \mid x, o_{k,<t})}{\pi_{\theta_{\text{old}}}(o_{k,t} \mid x, o_{k,<t})}, 1-\epsilon, 1+\epsilon\Big) \hat{A}_{k,t}(r) \Bigg) \Bigg],
\end{aligned} \quad (12)$$

where $\epsilon$ is a clipping hyperparameter and $\pi_\theta$ is the current policy model. Each $\mathcal{O}_k$ denotes a sampled token sequence from $\pi_{\theta_{\text{old}}}$, and $o_{k,t}$ represents the $t$-th token in the $k$-th sampled trajectory $\mathcal{O}_k$. For the input $x$, we sample a group of responses $\{\mathcal{O}_1, \mathcal{O}_2, \dots, \mathcal{O}_G\}$ for a given input $x$, and their rewards $\{r_1, r_2, \dots, r_G\}$ are obtained via the reward functions $r^{(1)}$ or $r^{(2)}$. The normalized advantage estimated score $\hat{A}_{k,t}(r)$ for each token is calculated as:

$$\hat{A}_{k,t}(r) = \frac{r_k - \text{mean}(\{r_1, r_2, \dots, r_G\})}{\text{std}(\{r_1, r_2, \dots, r_G\})}. \quad (13)$$

## 4 EXPERIMENTAL METHODOLOGY

This section describes the datasets, evaluation metrics, baselines, and implementation details.

**Datasets.** We first introduce the datasets used in our experiments, followed by the data statistics for golden reasoning trajectory construction.

Our experiments incorporate three QA scenarios: Text QA (2WikiMultihopQA (Ho et al., 2020)), Visual QA (InfoSeek (Chen et al., 2023b), Dyn-VQA (Li et al., 2024b), and WebQA (Chang et al., 2022)), and Table QA (Open-WikiTable (Kweon et al., 2023) and TabFact (Chen et al., 2019)). Specifically, we use 2WikiMultihopQA, InfoSeek, and Open-WikiTable for training and evaluation, while Dyn-VQA, TabFact, and WebQA are used for evaluation to assess the generalization capability of R1-Router. We adopt specialized retrievers tailored to each modality to support different QA tasks. For textual retrieval, we utilize BGE-M3 (Chen et al., 2024a) to retrieve relevant passages from the Wikipedia dump[1]. For multimodal scenarios, we employ UniIR (Wei et al., 2024) as a unified text-image retriever, retrieve related images from the M-BEIR corpus (Wei et al., 2024) and use corresponding image descriptions for augmentation. For table retrieval, we follow the setup of Open-WikiTable (Kweon et al., 2023), using the same dense retriever and table corpus in experiments. More details are shown in Appendix A.2.

To construct golden reasoning trajectories for training R1-Router, we employ R1-Distill-Qwen-32B (Guo et al., 2025) and Qwen2.5-VL-7B (Bai et al., 2025) to generate candidate reasoning paths. We then filter these reasoning trajectories by verifying whether they lead to correct answers. Further details on the data construction process are provided in Appendix A.3.

---

[1] https://dumps.wikimedia.org

Table 1: Overall Performance of R1-Router and Baselines (**best** and the second are highlighted).

| Method | KBs | In Distribution | | | Out of Distribution | | | Avg. |
|--------|-----|-----------------|---|---|---------------------|---|---|------|
| | | Open-WikiTable | 2WikiMQA | InfoSeek | Dyn-VQA | TabFact | WebQA | |
| **Vanilla Models** | | | | | | | | |
| Qwen2.5-VL-7B | - | 21.28 | 48.35 | 43.06 | 36.31 | 18.10 | 76.07 | 40.53 |
| R1-Distill-Qwen-32B | - | 22.75 | 51.78 | 37.20 | **39.98** | 19.10 | 79.41 | 41.70 |
| **RAG Methods w/o KB Routing** | | | | | | | | |
| Vanilla RAG | Text | 15.38 | 48.20 | 31.88 | 13.26 | 29.20 | 79.04 | 36.16 |
| Vanilla RAG | Image | 13.77 | 43.95 | 43.03 | 14.03 | 27.50 | 79.76 | 37.01 |
| Vanilla RAG | Table | 53.35 | 41.89 | 33.37 | 12.29 | 27.90 | 75.73 | 40.76 |
| Vanilla RAG | All | 49.99 | 48.31 | 39.06 | 14.12 | 34.90 | 76.63 | 43.84 |
| IRCoT | Text | 5.78 | 24.70 | 16.80 | 16.61 | 5.20 | 46.16 | 19.21 |
| IRCoT | Image | 5.32 | 18.94 | 27.47 | 18.04 | 2.10 | 47.06 | 19.82 |
| IRCoT | Table | 35.52 | 11.13 | 17.22 | 14.95 | 8.30 | 44.46 | 21.93 |
| IRCoT | All | 39.44 | 25.77 | 23.41 | 22.23 | 9.60 | 49.48 | 28.32 |
| IterRetGen | Text | 14.59 | 50.07 | 39.59 | 36.18 | 30.60 | 83.88 | 42.49 |
| IterRetGen | Image | 12.38 | 43.93 | 41.35 | 32.66 | 30.80 | 84.50 | 40.94 |
| IterRetGen | Table | 36.74 | 42.91 | 40.53 | 32.44 | 36.90 | 83.58 | 45.52 |
| IterRetGen | All | 38.95 | 50.99 | 40.94 | 36.08 | 38.60 | 84.19 | 48.29 |
| Search-O1 | Text | 9.72 | 28.12 | 18.52 | 16.40 | 29.60 | 16.78 | 19.86 |
| **RAG Methods w/ KB Routing** | | | | | | | | |
| CogPlanner | All | 16.50 | 49.28 | 42.23 | 36.60 | 33.10 | 84.82 | 43.76 |
| UniversalRAG | All | 31.12 | 47.30 | 37.25 | 11.91 | 26.00 | 79.48 | 38.84 |
| OmniSearch | All | 7.72 | 31.02 | 24.45 | 18.94 | 2.30 | 58.02 | 23.24 |
| MMSearch-R1[1] | All | 6.50 | 26.21 | 13.25 | 6.86 | 7.90 | 2.67 | 10.57 |
| MMSearch-R1[2] | All | 7.43 | 19.41 | 29.09 | 21.78 | 39.20 | 40.75 | 26.28 |
| R1-Router-3B | All | 53.85 | 55.18 | 37.45 | 37.58 | **52.60** | 89.54 | 54.37 |
| R1-Router-7B | All | **53.95** | **55.47** | **43.60** | 39.24 | 52.40 | **90.92** | **55.93** |

**Evaluation Metrics.** We utilize `F1-Recall` as our evaluation metric for all the tasks, which calculates the ratio of standard tokens between model responses and ground truth, following prior work (Li et al., 2024b).

**Baselines.** We compare R1-Router with several baseline methods, including vanilla (M)LLMs, vanilla RAG models, iterative RAG models, and RAG models with Knowledge Base (KB) routing. MMSearch-R1[1] uses the prompt provided in the paper, whereas MMSearch-R1[2] adopts our redesigned prompt. More implementation details are provided in Appendix A.8.

**Implementation Details.** We adopt Qwen2.5-VL-7B (Bai et al., 2025) as the backbone model for building our R1-Router model. To prevent infinite retrieval loops, we limit the number of retrieval iterations to at most 3 ($n \leq 3$). After knowledge accumulation, the initial query $q_0$ is re-fed into the model to produce the final answer through the $n + 1$-th reasoning step $\mathcal{R}_{n+1}$. More experimental details are shown in Appendix A.1

## 5 EVALUATION RESULTS

In this section, we first present the performance of R1-Router across various QA tasks, including Text QA, Visual QA, and Table QA. We then conduct ablation studies to examine the effectiveness of different training strategies. Next, we analyze how R1-Router performs adaptive retrieval during the reasoning process. Finally, we provide case studies to analyze the behavior of R1-Router.

### 5.1 OVERALL PERFORMANCE

The overall performance of R1-Router and baseline methods is shown in Table 1. For both vanilla and iterative RAG models, we evaluate two retrieval settings for evaluation: (1) retrieving 5 documents individually from each modality-specific KB (text, image, or table), and (2) aggregating all 15 documents retrieved from these KBs.

Overall, R1-Router consistently outperforms all baseline models, achieving an average performance gain of approximately 7%. Notably, R1-Router shows consistent improvements by adaptively routing queries to different KBs to collect information, highlighting its strong generalization ability and potential to serve as a universal solution for dealing with different QA tasks. Compared to

Table 2: Ablation Study.

| Method | In Distribution | | | Out of Distribution | | | Avg. |
|---|---|---|---|---|---|---|---|
| | Open-WikiTable | 2WikiMQA | InfoSeek | Dyn-VQA | TabFact | WebQA | |
| **R1-Router (Random Routing)** | | | | | | | |
| Prompt | 18.88 | 41.44 | 25.70 | 25.16 | 25.90 | 77.79 | 35.81 |
| SFT | 24.60 | 41.91 | 27.35 | 27.09 | 43.10 | 79.81 | 40.64 |
| Step-GRPO | 44.03 | 51.08 | 40.56 | 38.18 | 50.50 | 90.66 | 52.50 |
| **R1-Router** | | | | | | | |
| Prompt | 23.97 | 41.56 | 24.43 | 26.19 | 25.20 | 77.94 | 36.55 |
| SFT | 28.12 | 42.65 | 31.35 | 29.62 | 47.71 | 76.75 | 42.70 |
| GRPO | 10.41 | 28.56 | 24.18 | 17.23 | **54.10** | 11.94 | 24.40 |
| Step-GRPO | **53.95** | **55.47** | **43.60** | **39.24** | 52.40 | **90.92** | **55.93** |

RAG models without KB routing, R1-Router exhibits clear advantages by dynamically routing queries to different KBs for information retrieval. Within the group of RAG baselines lacking KB routing, incorporating evidence retrieved from multiple KBs leads to noticeable improvements in QA performance, indicating that different KBs contribute complementary information that supports MLLMs in answering queries. In addition, IterRetGen achieves substantially better performance than the vanilla RAG model, illustrating that iterative retrieval can help accumulate more related and sufficient information to answer these complex queries. While Search-O1 encourages Large Reasoning Models (LRMs) to perform adaptive retrieval during reasoning, it performs worse than other models. This may be attributed to the limitations of prompting-based methods in effectively guiding LRMs to utilize retrieval tools, a capability that may not have been sufficiently learned during pretraining (Jin et al., 2025). Furthermore, R1-Router outperforms RAG approaches equipped with KB-based routing, highlighting its effectiveness in extending the deep reasoning capabilities of MLLMs for QA tasks. Thrived on our Step-GRPO strategy, R1-Router jointly improves both routing and reasoning capabilities of MLLMs, leading to a more effective and adaptable RAG framework.

## 5.2 ABLATION STUDIES

This section conducts ablation studies to evaluate the effectiveness of different training strategies, including Prompt, SFT, GRPO, and Step-GRPO. To assess the contribution of KB routing, we further compare two evaluation settings: (1) random routing, where a KB is selected at random for each intermediate query, and (2) self-routing, where the MLLM itself determines the most relevant KB.

As shown in Table 2, R1-Router (Step-GRPO) consistently outperforms all baselines, yielding over 10% improvements across all tasks. This result highlights the effectiveness of Step-GRPO in guiding the MLLM through multi-step reasoning and structured knowledge acquisition. R1-Router (SFT) outperforms prompt-based methods overall, but exhibits limited effectiveness on the Open-2WikiTable and WebQA datasets. This suggests that relying solely on SFT for model optimization hinders the ability of MLLMs to generalize across diverse QA scenarios. However, for R1-Router (GRPO), reward sparsity and the lack of action continuity prevent the model from learning effective reasoning, thereby degrading performance. In contrast, R1-Router (Step-GRPO) yields more consistent and significant improvements across all tasks, demonstrating its superior ability to accumulate knowledge through RL optimization. Notably, even when combined with random routing, R1-Router (Step-GRPO) maintains robust performance gains over Prompt and SFT methods, suggesting that its benefits stem not only from routing precision but also from more effective reasoning and intermediate query generation. By further incorporating self-routing, R1-Router (Step-GRPO) achieves an additional 3% improvement. This confirms the effectiveness of R1-Router in dynamically selecting relevant KBs, producing more accurate answers.

## 5.3 ANALYZING THE BEHAVIOR OF R1-ROUTER IN ADAPTIVE RETRIEVAL AND REASONING

In this section, we first examine the effectiveness of Step-GRPO in enabling MLLMs to perform adaptive reasoning and retrieval. We then analyze the KB routing behaviors across different models to understand their knowledge-seeking strategies better.

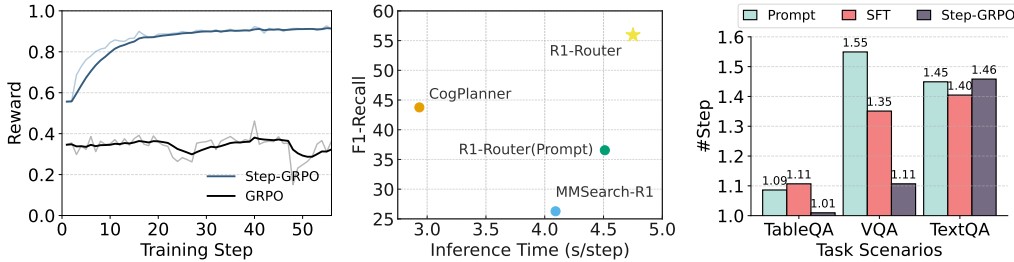

(a) Evolution of Reward During GRPO and Step-GRPO Training.

(b) Computational Cost with Other KB Routing Baselines.

(c) Reasoning Steps Required for Correct Question Answering.

Figure 2: Performance of Step-GRPO.

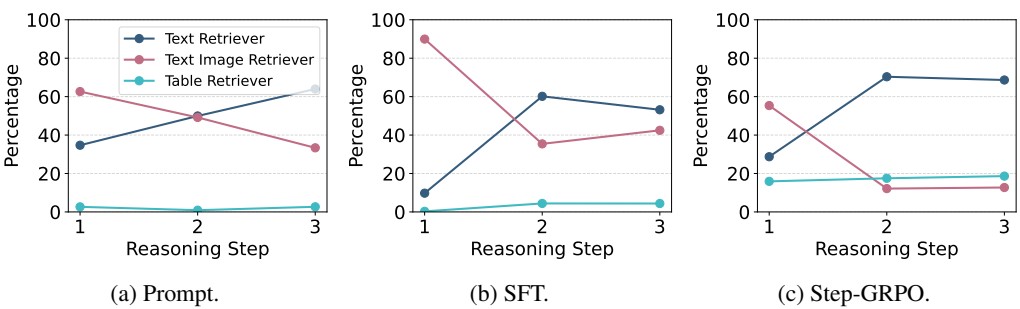

(a) Prompt.

(b) SFT.

(c) Step-GRPO.

Figure 3: Query Routing Performance on VQA During the Reasoning Process of R1-Router. Three training strategies are compared, including Prompt, SFT, and Step-GRPO.

**Effectiveness of Step-GRPO.** As illustrated in Figure 2, we evaluate the impact of our Step-GRPO training method on improving MLLMs' reasoning capabilities. In Figure 2a, we show the progression of reward scores during training. Training with GRPO remains unstable under discontinuous data and sparse rewards. Step-GRPO remedies this via unified optimization of contiguous actions and fine-grained rewards, leading to stable growth. As shown in Figure 2b, across KB-routing baselines, the R1-Router delivers more than a 12% gain in F1-Recall with under 2s additional per-step latency, aligning well with the performance–computational-cost trade-off. More computational cost details can be seen in Appendix A.7. Figure 2c presents the average number of reasoning steps required to answer queries correctly. We categorize all test data into three types–Table QA, Visual QA (VQA), and Text QA–based on the kind of knowledge needed. In the TextQA scenario, Step-GRPO performs comparably to other training strategies in terms of average reasoning steps, indicating that deeper reasoning alone may not reduce retrieval complexity for purely textual queries. In contrast, in both VQA and Table QA scenarios, R1-Router significantly reduces the number of reasoning steps, demonstrating its effectiveness in guiding MLLMs to route queries more efficiently across different KBs for conducting more effective knowledge usage.

**Routing Preferences of R1-Router During Reasoning.** We further investigate how Step-GRPO reduces reasoning steps by analyzing the evolution of routing preferences throughout the reasoning process of R1-Router. As shown in Figure 3, we compute the proportion of selected KBs at each reasoning step on the VQA task. Compared with R1-Router (Prompt), the R1-Router (SFT) model demonstrates a rigid retrieval pattern, inherited mainly from dataset-specific biases. It tends first to use the Text-Image Retriever to obtain relevant visual descriptions from the predefined task-specific KB. Then, it alternates between the Text-Image Retriever and the Text Retriever to search for multiple KBs. In contrast, Step-GRPO encourages more adaptive and flexible routing behavior. Specifically, R1-Router (Step-GRPO) increasingly shifts its retrieval preference from the Text-Image Retriever to the Text Retriever, indicating that textual knowledge (e.g., from Wikipedia) often plays a more crucial role in answering visual queries. Moreover, it selectively employs the Table Retriever when appropriate, demonstrating its ability to leverage multimodal KBs based on the information need adaptively. This adaptive knowledge-seeking behavior highlights the potential of Step-GRPO in enabling more generalizable and flexible RAG systems for multimodal QA tasks.

```
Question: What is the closest parent taxonomy of this bird?
Ground Truth: Setophaga

Exploration Trajectory of R1-Router
<think> To determine the closest parent taxonomy of a bird, we need to identify the species of the bird shown in the image.
This requires a visual analysis of the bird, which can be achieved using a Text Image Retriever. </think>
<sub-question> What is the name of the bird shown in the image? </sub-question>
<ret> Text Image Retriever </ret>

Retrieved Documents:
tir-1: Dendroica-castanea-001 Bay-breasted Warbler. Canada Rondeau Provincial Park, Ontario, Canada. tir-2: Icterus galbula-
Baltimore, Maryland, USA -juvenile-8 (1) A juvenile Baltimore Oriole inthe grounds of Maryland Zoo, Baltimore, Maryland, USA.
tir-3: Poospiza lateralis -Extrema, Minas Gerais,Brazil-8 A Buff-throated Warbling Finch in Extrema, Minas Gerais, Brazil.
<answer> The name of the bird shown in the image is most likely bay-breasted warbler (Dendroica castanea). This conclusion is
based on the description of the bay-breasted warbler in the text retriever information (tir-1), which mentions a bird with a
distinctive chest pattern, which could be characteristic of the bird shown in the image. The image appears to be of a bird with
a similar pattern on its chest, which is consistent with the description of the bay-breasted warbler. </answer>

<think> The next step is to determine the closest parent taxonomy of the Bay-breasted Warbler. This requires identifying the
appropriate taxonomic classification for the bird. </think>
<sub-question> What is the closest parent taxonomy of the Bay-breasted Warbler? </sub-question>
<ret> Text Retriever </ret>

Retrieved Documents:
ter-1: Bay-breasted Warbler The Bay-breasted Warbler (Setophaga castanea) is a small species of songbird in the New World
warbler family, Parulidae. It is one of thirty-four species in the diverse genus Setophaga. ter-2: Bay-breasted warbler
Warblers may look very similar to the blackpoll warbler (Setophaga striata). The genus name Setophaga is from Ancient Greek ses,
moth, and, eating, and ...
<answer> ... is one of thirty-four species in the diverse genus Setophaga. This classification places the Bay-breasted Warbler
in the order Passeriformes ... </answer>
...
<think> The provided answers all suggest that the closest parent taxonomy of the Bay-breasted Warbler, which the image appears
to be of, is within the order Passeriformes... </think>
<answer> ... and is one of thirty-four species in the diverse genus Setophaga. </answer>
```

Figure 4: An Example from the Visual QA Task Demonstrating the Performance of R1-Router.

## 5.4 CASE STUDY

To further demonstrate the effectiveness of R1-Router, we present a case study from the InfoSeek dataset on a VQA task. As illustrated in Figure 4, we show the step-by-step reasoning process of R1-Router for a query that asks for the closest parent taxonomy of the bird depicted in the image.

In the first reasoning step, R1-Router initiates a deliberation process and decomposes the task into a subgoal: conducting a visual analysis. It formulates an intermediate query–"What is the name of the bird shown in the image?"–and selects the Text-Image Retriever to obtain relevant visual information. By leveraging this retriever, R1-Router retrieves image descriptions of semantically similar images from the knowledge base, which include key entities such as "Bay-breasted Warbler", that describe the bird in the given image. Notably, R1-Router performs a reflection step to verify that the bird is indeed a Bay-breasted Warbler, based on distinctive visual features (e.g., chest patterns) mentioned in the retrieved evidence. This demonstrates the effectiveness of R1-style deep reasoning in RAG-based modeling. In the second step, R1-Router formulates a follow-up query–"What is the closest parent taxonomy of the bay-breasted warbler?"–and selects the Text Retriever to collect factual knowledge. Using the retrieved content, R1-Router identifies the genus of the Bay-breasted Warbler as "Setophaga" and ultimately outputs the correct final answer. This case study demonstrates the ability of R1-Router to dynamically decide when and from which knowledge source to retrieve information during problem-solving, showcasing its strong reasoning and retrieval planning capabilities.

## 6 CONCLUSION

This paper proposes R1-Router, a method designed to dynamically determine when and where to retrieve relevant knowledge during reasoning. Specifically, R1-Router introduces Stepwise Group Relative Policy Optimization (Step-GRPO). This optimization algorithm computes step-specific rewards at intermediate reasoning steps to guide MLLMs to learn how to retrieve information from different knowledge bases during reasoning. Experimental results show that R1-Router can conduct practical retrieval actions across multiple knowledge bases and perform iterative reasoning conditioned on the current problem-solving context. This enables it to effectively tackle complex queries requiring multi-step reasoning and heterogeneous information sources. Moreover, R1-Router provides a promising step toward building generalizable RAG systems by enabling MLLMs to effectively manage multimodal knowledge bases and handle a broad range of query types.

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

## A APPENDIX

### A.1 IMPLEMENTATION DETAILS

More training hyperparameters are listed in Table 3.

Table 3: Training Details.

| Parameter | Value |
|---|---|
| Rollout Batch Size | 200 |
| Max Grad Norm | 1.0 |
| Learning Rate | $1.0 \times 10^{-6}$ |
| Weight Decay | $1.0 \times 10^{-2}$ |
| Temperature | 1.0 |
| Tensor Parallel Size | 2 |
| Rollout Group | 8 |
| Seed | 42 |
| $\epsilon$ | 0.2 |
| $\alpha, \beta$ | 0.5 |

## A.2 MORE EXPERIMENTAL DETAILS OF KNOWLEDGE BASES USING IN R1-ROUTER

**Retrievers.** We employ BGE-M3 (Chen et al., 2024a) as our text retriever when R1-Router routing queries to the textual knowledge base. It utilizes textual queries to retrieve factual knowledge from the textual knowledge base, aiding MLLMs in generating more accurate answers. UniIR (Wei et al., 2024) is used as our text-image retriever to obtain visual descriptions based on the query text and the image. Specifically, UniIR independently encodes the query text and image, fuses their features into a unified multimodal representation, and retrieves the most relevant text-image pairs from the corpus. The textual components of these pairs are then used as our retrieved evidence. For the table knowledge base, we follow the Open-WikiTable (Kweon et al., 2023) and use the same table retriever to search for relevant tables based on queries. Retrieval cases from different retrievers are shown in Figure 5.

**Retrieval Corpus.** For the text corpus, we use the English Wikipedia dump `20241020` as the source. Following prior work (Chen et al., 2017; Wang et al., 2019), we extract the textual content from the dump and segment each article into multiple disjoint text blocks of 100 words each, which serve as the basic retrieval units. To ensure the quality of the text KB, we discard any text blocks containing fewer than 7 words, yielding a final collection of 52,493,415 passages. Each passage is further prepended with the title of the Wikipedia article from which it originates. For the text-image corpus, we follow the setting of UniIR (Wei et al., 2024) and adopt M-BEIR as our image description corpus, which contains over 5.6 million candidates. For the table corpus, we follow Open-WikiTable (Kweon et al., 2023) and utilize their corpus as our table corpus, which includes 24,680 tables.

## A.3 DATA CONSTRUCTION OF GOLDEN REASONING TRAJECTORIES

To efficiently train and develop R1-Router, we construct a QA dataset with golden reasoning trajectories across different task scenarios. This subsection details the construction process.

**Data Sources.** We constructed the query-answer pairs by collecting samples from existing QA datasets spanning diverse task scenarios. Specifically, we used 2WikiMultihopQA (Ho et al., 2020) for Text QA, InfoSeek (Chen et al., 2023b) for Visual QA (VQA), and Open-WikiTable (Kweon et al., 2023) for Table QA. These datasets contain complex queries that often require multiple steps to be answered accurately.

**Data Synthesis.** We built an automated data pipeline to synthesize golden reasoning trajectories $\mathcal{R}^*$ for training R1-Router. Specifically, we prompt LRMs or MLLMs to generate step-by-step reasoning trajectories iteratively, applying rejection sampling to filter and retain high-quality examples. The process of intermediate query generation and retriever identifier selection can be expressed as:

$$(q_i, s_i) = \mathcal{M}\left(q_0, \mathcal{R}_{1:i-1}\right), \tag{14}$$

where $q_0$ represents the initial query, $\mathcal{R}_{1:i-1}$ represents previous reasoning trajectories, $q_i$ and $s_i$ denote the intermediate query and the selected retriever identifier in the reasoning step $i$. $\mathcal{M}$ denotes the LRM or MLLM used in our pipeline. For VQA tasks, we adopt Qwen2.5-VL-7B (Bai et al., 2025) as the foundation model, while for Table QA and Text QA tasks, we employ R1-Distill-Qwen-32B (Guo et al., 2025). Then the intermediate answer generation steps can be expressed

as:

$$a_i = \mathcal{M}(q_i, d_i), \tag{15}$$

where $d_i$ represents the retrieved content, and $a_i$ denotes the corresponding intermediate answer. The final answer generation step is formatted as:

$$a_{n+1} = \mathcal{M}\left(q_0, \mathcal{R}_{1:n}\right), \tag{16}$$

where $a_{n+1}$ is the final answer generated. To ensure the quality of training data, we applied rejection sampling to filter the data, retaining only those reasoning paths whose answers achieve an accuracy of 1 respect to the ground-truth answer $a_{n+1}^*$ as the ground-truth reasoning steps $\mathcal{R}^*$, which is formatted as:

$$\mathcal{R}^* = \underbrace{\{(q_1, s_1), a_1\}}_{\mathcal{R}_1^*}, \ldots, \underbrace{\{(q_n, s_n), a_n\}}_{\mathcal{R}_n^*}, \underbrace{\{q_0, a_{n+1}^*\}}_{\mathcal{R}_{n+1}^*}\Big|_{\mathrm{Acc}(a_{n+1}, a_{n+1}^*)=1}, \tag{17}$$

thus providing a high-quality step-by-step training dataset for R1-Router.

### A.4 More Details on the Fine-grained Reward

**Format Reward.** We set the format reward to ensure the generated content is enclosed in special tokens. Specifically, the reasoning process of R1-Router, intermediate queries and invoked retrievers should be enclosed within the **<think>...</think>**, **<sub-question>...</sub-question>** and **<ret>...</ret>** tags, respectively. Based on the above format requirements, the format reward in this step can be defined as:

$$r_{\mathrm{format}}(q_i, s_i) = \begin{cases} 1, & \text{if the format of } q_i \text{ and } s_i \text{ are correct} \\ 0, & \text{otherwise} \end{cases} \tag{18}$$

where $q_i$ and $s_i$ are the intermediate query and the identifier of the selected retriever at the $i$-th reasoning step. When generating the intermediate and final answer, we define the correct format of the reasoning process and answer should be enclosed within the **<think>...</think>** and **<answer>...</answer>** tags, respectively. Based on the above format requirements, the format reward in this step can be defined as:

$$r_{\mathrm{format}}(a_i) = \begin{cases} 1, & \text{if the format of } a_i \text{ is correct} \\ 0, & \text{otherwise} \end{cases} \tag{19}$$

where $a_i$ is the generated answer at the $i$-th reasoning step.

**Query Reward.** We define the query reward as the textual similarity between the generated intermediate query and the ground-truth intermediate query. Specifically, we employ the BGE-M3-embedding (Chen et al., 2024a) model as the encoder $E(\cdot)$ to project them into a high-dimensional embedding space and compute their similarity, which can be expressed as:

$$r_{\mathrm{ask}}(q_i) = sim\left(E(q_i), E(q_i^*)\right), \tag{20}$$

where $q_i$ is the generated query at the $i$-th reasoning step and $q_i^*$ is the ground truth query. The *sim* denotes the similarity function, where we use the dot product to calculate the similarity between $q_i$ and $q_i^*$.

**Routing Reward.** We utilize the accuracy of knowledge bases routing as our routing reward, which is calculated as follows:

$$r_{\mathrm{route}}(s_i) = \begin{cases} 1, & \text{if } s_i = s_i^* \\ 0, & \text{otherwise} \end{cases} \tag{21}$$

where $s_i$ is the selected retriever identifier at the $i$-th reasoning step and $s_i^*$ is the ground truth retriever identifier.

**Answer Reward.** We adopt F1-Recall (Li et al., 2024b) to compute the answer reward for intermediate answers $a_i$ ($1 \leq i \leq n$) at the $i$-th reasoning step, and accuracy to evaluate the final answer $a_{n+1}$. Specifically, the answer reward for intermediate answers $a_i$ ($1 \leq i \leq n$) can be expressed as:

$$r_{\mathrm{answer}}(a_i) = \text{F1-Recall}\left(a_i, a_i^*\right), \tag{22}$$

Table 4: Overall Performance of R1-Router and Baselines based on 3B Parameter (**best** are highlighted).

| Method | KBs | In Distribution | | | Out of Distribution | | | Avg. |
|--------|-----|-----------------|--|--|---------------------|--|--|------|
| | | Open-WikiTable | 2WikiMQA | InfoSeek | Dyn-VQA | TabFact | WebQA | |
| Qwen2.5-VL-3B | - | 18.62 | 39.66 | 33.92 | 30.10 | 21.80 | 61.70 | 34.30 |
| Vanilla RAG | All | 44.57 | 47.58 | 25.95 | 12.22 | 41.20 | 82.08 | 42.27 |
| IterRetGen | All | 33.37 | 46.63 | 35.83 | 33.02 | 45.90 | 82.30 | 46.18 |
| IRCoT | All | 35.15 | 34.68 | 24.36 | 23.83 | 35.20 | 71.67 | 37.48 |
| CogPlanner | All | 14.18 | 45.29 | 36.78 | 31.48 | 41.60 | 83.13 | 42.08 |
| UniversalRAG | All | 17.85 | 46.08 | 34.61 | 10.86 | 33.80 | 81.81 | 37.50 |
| R1-Router-3B | All | **53.85** | **55.18** | **37.45** | **37.58** | **52.60** | **89.54** | **54.37** |

Table 5: Computational cost details of KB routing baselines, where "FR" means *F1-Recall*.

| Method | Open-WikiTable | | 2WikiMQA | | InfoSeek | | Dyn-VQA | | TabFact | | WebQA | | Avg. | |
|--------|------|------|------|------|------|------|------|------|------|------|------|------|------|------|
| | FR | Time | FR | Time | FR | Time | FR | Time | FR | Time | FR | Time | FR | Time |
| CogPlanner | 16.50 | 2.962 | 49.28 | 1.796 | 42.23 | 4.747 | 36.60 | 2.518 | 33.10 | 1.209 | 84.82 | 3.533 | 43.76 | 2.934 |
| MMSearch-R1[2] | 7.43 | 4.419 | 19.41 | 3.652 | 29.09 | 3.639 | 21.78 | 3.690 | 39.20 | 4.047 | 40.75 | 3.941 | 26.28 | 4.092 |
| R1-Router(Prompt) | 23.97 | 3.801 | 41.56 | 2.312 | 24.43 | 5.239 | 26.19 | 4.101 | 25.20 | 3.653 | 77.94 | 6.674 | 36.55 | 4.511 |
| R1-Router(SFT) | 28.12 | 12.161 | 42.65 | 7.537 | 31.25 | 19.614 | 29.62 | 10.916 | 47.71 | 11.235 | 76.75 | 13.921 | 42.70 | 13.191 |
| R1-Router | 53.95 | 5.569 | 55.47 | 3.049 | 43.60 | 5.895 | 39.24 | 3.284 | 52.40 | 4.338 | 90.92 | 5.012 | 55.93 | 4.751 |

where $a_i^*$ represents the ground-truth intermediate answer at the $i$-th reasoning step. The F1-Recall can be calculated as:

$$\text{F1-Recall}\,(a_i, a_i^*) = \frac{|a_i \cap a_i^*|}{|a_i^*|}, \tag{23}$$

where $|a_i \cap a_i^*|$ indicates the number of words in the intersection between the predicted answer $a_i$ and the ground-truth answer $a_i^*$. For the predicted final answer $a_{n+1}$, the answer reward can be calculated using the Accuracy score:

$$r_{\text{answer}}(a_{n+1}) = \text{Acc}\,(a_{n+1}, a_{n+1}^*), \tag{24}$$

where $a_{n+1}^*$ indicates the ground truth of the final answer. The Accuracy score is usually used to evaluate the performance of QA systems in existing works (Asai et al., 2023; Li et al., 2024a).

## A.5 PROMPT TEMPLATES USED IN R1-ROUTER

The 8-shot prompt templates used by the R1-Router (Prompt) and R1-Router (SFT) methods for intermediate query generation and retriever selection across three task scenarios are shown in Figure 6, Figure 7, and Figure 8. The same prompt templates are applied during data construction, with samples containing incorrect final answers subsequently filtered out. We show the prompt templates of R1-Router (Step-GRPO) for query generation and routing in Figure 9. The prompt templates used by all R1-Router models for intermediate and final answer generation are shown in Figure 10 and Figure 11, respectively.

## A.6 MORE EXPERIMENTS BASED ON 3B PARAMETER

Table 4 shows the overall performance of R1-Router and Baselines based on Qwen2.5-VL-3B. The training strategy of R1-Router delivers substantial gains even at this small scale.

## A.7 MORE COMPUTATIONAL COST DETAILS OF THE KB ROUTING BASELINES

Table 5 lists the computational-cost details for the KB-routing baselines, showing that R1-Router achieves a favorable performance–cost trade-off.

## A.8 MORE IMPLEMENTATION DETAILS OF THE BASELINE METHODS

In this section, we introduce the implementation details of the baseline methods used in our experiments.

**Vanilla Models.** For vanilla (M)LLM modeling, the model directly receives the query along with the image and generates answers without any retrieval process. In contrast, for LLMs without multimodal capabilities, the input image is first converted into a detailed textual description via image captioning provided by MLLMs. We prompt Qwen2.5-VL-7B and R1-Distill-Qwen-32B to directly answer the initial query. For R1-Distill-Qwen-32B, we utilize Qwen2.5-VL-7B to produce detailed textual descriptions of the image and incorporate them into the input context to facilitate answer generation. The prompt templates used are shown in Figure 12.

**Vanilla RAG.** The vanilla RAG model incorporates retrieved evidence as additional context to assist MLLMs in answering queries (Sharifymoghaddam et al., 2024; Ram et al., 2023). We use Qwen2.5-VL-7B as the foundation model to implement the Vanilla RAG methods and retrieve relevant knowledge from specific knowledge bases, incorporating the top-5 retrieved evidence pieces into the input context. We further concatenate these evidence pieces retrieved from multiple knowledge bases to expand the input context and evaluate the model's ability to leverage information from diverse knowledge bases. The prompt template used for Vanilla RAG is illustrated in Figure 12.

**IRCoT.** IRCoT generates a chain-of-thought (CoT) (Wei et al., 2022) as the follow-up queries. We follow the paper (Trivedi et al., 2022) to implement IRCoT as our baseline method. IRCoT iteratively guides the model to generate a Chain-of-Thought (CoT) (Wei et al., 2022) after each retrieval step, using the final sentence of the CoT as the query for the next-turn retrieval until the CoT includes "The answer is" or reaches the maximum number of retrieval iterations. Then the final sentence of the CoT is considered the final answer. The 8-shot prompt modeling method we used to implement IRCoT is shown in Figure 13. The maximum number of iterations is set to 3.

**CogPlanner.** CogPlanner (Yu et al., 2025) prompts Qwen2.5-VL-7B to iteratively generate intermediate queries, decide whether retrieval is necessary, and select the appropriate knowledge base for retrieval. The iteration process terminates when the model determines that the accumulated information is sufficiently comprehensive and the current query is adequately clear. Subsequently, the model integrates all retrieved evidence to produce the final answer. The prompt templates used by CogPlanner are illustrated in Figure 14 and Figure 15.

**IterRetGen.** We adopt the Iterative Retrieve-and-Generate (IterRetGen) (Shao et al., 2023) framework as our baseline. IterRetGen alternates between iterative retrieval and generation and uses the retrieved evidence and the current query to generate the new intermediate query for the next iteration. This iterative process continues until the model outputs "None" as the intermediate query or the maximum number of iterations is reached. The prompt template used for intermediate query generation is shown in Figure 15. We use Qwen2.5-VL-7B as the foundation model. The prompt templates used for intermediate answer generation and final answer generation are shown in Figure 16 and Figure 17. The maximum number of iterations is set to 3.

**Search-O1.** We implement Search-O1(Li et al., 2025a) based on their official codebase[2] and adopt R1-Distill-Qwen-7B (Guo et al., 2025) as the backbone reasoning model. When the model generates the special tokens `<|begin_search_query|>` and `<|end_search_query|>`, Search-O1 performs retrieval from a textual knowledge base using the generated query. The top-5 retrieved evidence pieces are then refined and incorporated into the reasoning chain using the tokens `<|begin_search_result|>` and `<|end_search_result|>`. This iterative retrieval and reasoning process continues until the model produces a final answer or the maximum number of reasoning steps is reached, which is set to 3.

**UniversalRAG.** We use the 8-shot prompt from their paper to implement UniversalRAG (Yeo et al., 2025), guiding Qwen2.5-VL-7B to either directly answer the query or route to an appropriate knowledge base for retrieval. Once a knowledge base is selected, it formulates a search query to retrieve the top 5 relevant documents and uses them to generate the final answer. The retrieval setting is consistent with that of the R1-Router and the prompt template of UniversalRAG is shown in Figure 18.

**OmniSearch.** We implement OmniSearch (Li et al., 2024b) using their official checkpoints[3] and the prompt templates provided in their paper. The model is trained based on Qwen-VL-Chat via Supervised Fine-Tuning (SFT) on GPT-4V-generated synthetic data to enable MLLMs to iteratively

---

[2]https://github.com/sunnynexus/Search-o1
[3]https://huggingface.co/Alibaba-NLP/OmniSearch-Qwen-VL-Chat-en

generate intermediate queries and select appropriate knowledge bases to retrieve. The iterative process continues until the model outputs a "Final Answer" or reaches the maximum number of iterations. The prompts used for OmniSearch are shown in Figure 19.

**MMSearch-R1.** We implement MMSearch-R1 (Wu et al., 2025a) using their official checkpoint[4], the prompt templates of MMSearch-R1[1] provided in their paper, as shown in the Figure 20, Figure 21 and Figure 22, which force the model to use image retriever first, neglecting the diversity needs of different queries, and the prompt templates of MMSearch-R1[2] are redesigned as shown in the Figure 22, Figure 23, and Figure 24. The model is trained based on Qwen2.5-VL-7B via GRPO to enable MLLMs to determine whether to retrieve and to choose appropriate knowledge bases to retrieve.

### A.9 ADDTIONAL CASE STUDIES OF R1-ROUTER

In this subsection, we present additional case studies to further demonstrate the effectiveness of R1-Router. Specifically, we showcase examples from various task scenarios to illustrate how R1-Router adaptively plans intermediate queries, selects appropriate knowledge bases for retrieval, and integrates retrieved information to construct coherent reasoning trajectories.

We begin with the case shown in Figure 25, which is drawn from the Open-WikiTable QA dataset under the Table QA task. The example queries the length of Dell Curry's tenure on the Toronto Raptors all-time C roster. R1-Router determines that no intermediate queries are necessary and directly employs the Table Retriever to obtain the relevant information. This highlights R1-Router's capability to dynamically decide when to stop intermediate query generation and generate the final answer. Furthermore, in the second step, R1-Router accurately performs the reflection mechanism by verifying that Dell Curry's tenure lasted four years (1999–2002), ensuring consistency across sub-questions and corresponding answers.

We next analyze the case illustrated in Figure 26, which comes from the Dyn-VQA dataset and belongs to the VQA task. The initial query for this case is: "What city was she born in?", accompanied by the corresponding image. R1-Router uses the Text Image Retriever to obtain the image description to identify that the person in the image is "Emily Dickinson". Then, it formulates a follow-up query "Where was Emily Dickinson born?", and uses the textual knowledge base to supplement factual knowledge for problem-solving. Based on retrieved content, R1-Router identifies that the birth city of Emily Dickinson is Amherst, Massachusetts. Notably, R1-Router ceases generating further intermediate queries once the original question can be answered with the accumulated knowledge, demonstrating its ability to adaptively determine when sufficient information has been gathered for effective reasoning.

The final case, illustrated in Figure 27, is drawn from the 2WikiMultihopQA dataset, which belongs to the Text QA task. The question asks about the place of death of Anastasia of Serbia's husband. R1-Router initially decides to clarify "Who was Anastasia of Serbia's husband?" and selects the Text Retriever to obtain relevant information. Upon confirming that Anastasia of Serbia's husband was Stefan Nemanja, R1-Router proceeds with the follow-up query, "Where did Stefan Nemanja die?", and again utilizes the text retriever to gather the necessary knowledge. Then the R1-Router got the final answer "Hilandar Monastery" with the accumulated information. This case demonstrates that R1-Router is capable of generating high-quality subquestions that effectively decompose the original query and guide the reasoning process.

---

[4]https://huggingface.co/lmms-lab/MMSearch-R1-7B

**A Retrieval Case from Text Retriever**

**Question:** Where was the place of death of Tamatoa Vi's father?
**Ground Truth:** Huahine

**Top-3 Retrieved Documents From Text Retriever(ter)**
**ter-1:** Tamatoa VI Ra'iatea and Taha'a in 1884, but was deposed in 1888. Biography. Tamatoa VI was born as Ioata Ariimate Teururai a Teururai and died as Teururai Teuhe a Teururai. He was the last King of Ra'iatea and Taha'a. He was the second son of King Teururai, Ariimate of Huahine. His mother, Princess Tehaapapa Maerehia of Raiatea, was the only living child of King Tamatoa IV of Raiatea. She became Queen regnant of Huahine under the regnal name of Tehaapapa II in july 8th 1868 after her husband was deposed by the chiefs in the night of july 7th 1868.
**ter-2:** Tamatoa VI the two islands of Ra'iatea and Taha'a on March 16, 1888. After his abdication, he returned to his natal island where he was later proclaimed chief of the district of Tefarerii in the name of Ioata, the same year that his niece Queen Tehaapapa III abdicated. Marriage and children. He married Tetuanuimarama a Atitioroi (daughter of Atitioroi a Ori, of the Tati family of Papara) and had three sons and three daughters: Their children remain the pretenders to the royal family of Raiatea and Taha'a since the end of the monarchy on this island. He died at Huahine in 1905.
**ter-3:** Tamatoa VI Tamatoa VI Prince Ariimate Teururai, later known as King Tamatoa VI (7 June 1853 – 15 September 1905), was a member of a Tahitian royal family, the \"House of Teururai\" which reigned on the Tahitian island of Huahiné and Maia'o during the 19th century. He was installed as king of Ra'iatea and Taha'a in 1884, but was deposed in 1888. Biography. Tamatoa VI was born as Ioata Ariimate Teururai a Teururai and died as Teururai Teuhe a Teururai. He was the last King of Ra'iatea and Taha'a. He was the second son of King Teururai, Ariimate of Huahine.

**A Retrieval Case from Text Image Retriever**

**Question:** Which body of water is this building located in or next to?
**Ground Truth:** Lake Superior

**Top-3 Retrieved Documents From Text Image Retriever(tir)**
**tir-1:** Split Rock Lighthouse on Lake Superior's North Shore. The lighthouse and grounds are a Minnesota state park 9 miles Southeast of Silver Bay.
**tir-2:** Split Rock Lighthouse - Two Harbors, Minnesota.
**tir-3:** Day Hiking Trails: Explore Split Rock Lighthouse via day hike.

**A Retrieval Case from Table Retriever**

**Question:** From the list of the main characters of CSI: NY, which actors appeared first in episode "Blink" 1, 2, 3?
**Ground Truth:** Gary Sinise

**Top-3 Retrieved Documents From Table Retriever(tar)**
**tar-1:** [Title]List of CSI: NY characters[Section title]Main characters[Caption]Main characters[Table name] table_1 [Header]Character_[sep]Portrayed_by[sep]First_appearance[sep]Last_appearance[sep]Duration[sep]Episodes[Rows][Row]Dr. Sheldon Hawkes CSI[sep]Hill Harper[sep]Blink 1[sep]Today is Life[sep]1.01–9.17[sep]197[Row]Don Flack Homicide Detective[sep]Eddie Cahill[sep]Blink[sep]Today is Life[sep]1.01–9.17[sep]197[Row]Aiden Burn CSI Detective[sep]Vanessa Ferlito[sep]Blink 1[sep]Heroes[sep]1.01–2.02, 2.23[sep]26
**tar-2:** [Title]List of CSI: NY characters[Section title]Notable villains[Caption]Notable villains[Table name]table_2 [Header]Character_[sep]Portrayed_by[sep]Crime[sep]First_appearance[sep]Last_appearance[Rows][Row]Sonny Sassone[sep] Michael DeLuise [sep]Murder (2 counts)[sep]Tanglewood[sep]Run Silent, Run Deep[Row]Frankie Mala[sep]Ed Quinn[sep]Attempted murder (Attacked Stella)[sep]Grand Murder at Central Station[sep]All Access[Row]Henry Darius[sep]James Badge Dale[sep]Murder (15 counts)[sep] Felony Flight ( CSI: Miami crossover)[sep]Manhattan Manhunt[Row]D.J. Pratt[sep]Chad Williams[sep]Murder / rape (1 / 2 counts) (Killed Aiden)[sep]Summer In The City[sep]Heroes[Row]Shane Casey[sep]Edward Furlong[sep]Murder (8 counts)[sep]Hung Out to Dry[sep]The 34th Floor[Row]Clay Dobson[sep]Joey Lawrence[sep]Murder (3 counts)[sep]Past Imperfect[sep]Comes Around
**tar-3:** [Title]List of CSI: NY characters[Section title]Main characters[Caption]Main characters[Table name]table_1_11240028_1[Header]Character_[sep]Portrayed_by[sep]First_appearance[sep]Last_appearance[sep]Duration[sep]Episodes[Rows][Row]Mac Taylor CSI Detective[sep]Gary Sinise[sep]Blink 1, 2, 3[sep]Today is Life[sep]1.01–9.17[sep]197[Row]Jo Danville CSI Detective[sep]Sela Ward[sep]The 34th Floor[sep]Today is Life[sep]7.01–9.17[sep]57[Row]Danny Messer CSI Detective[sep]Carmine Giovinazzo[sep]Blink 1[sep]Today is Life[sep]1.01–9.17[sep]197[Row]Lindsay Monroe Messer CSI Detective[sep]Anna Belknap[sep]Zoo York[sep]Today is Life[sep]2.03–9.17[sep]172 4[Row]Dr. Sid Hammerback Chief Medical Examiner[sep]Robert Joy[sep]Dancing with the Fishes[sep]Today is Life[sep]2.05–9.17[sep]168 4[Row]Adam Ross Lab Technician[sep]A. J. Buckley[sep]Bad Beat[sep]Today is Life[sep]2.08–9.17[sep]141 4

Figure 5: Case Studies of Different Retrievers.

```
You are a professional question decomposition expert for multi-hop QA systems. Your task is to decompose complex questions into
strictly single-hop sub-questions and select appropriate retrievers.

Strict Output Format:
<think>[Analyze the original question and determine the next required sub-question. Do NOT reveal answers or perform multi-hop
reasoning.]</think><sub-question>[Exactly ONE single-hop question one time. If no further information is needed to answer the
origin question, write 'None'.]</sub-question><ret> [Choose 1 retriever from: Text Retriever, Text Image Retriever, Table
Retriever. Write 'None' if <sub-question> is 'None'.] </ret>

Critical Rules:
1.Atomic Sub-question Definition:
   - A sub-question is "atomic" only if:
      a) It cannot be further decomposed into simpler questions
      b) It requires exactly **one retrieval action** to answer
      c) Does NOT depend on answers to previous sub-questions
      d) It can be helpful to answer the origin question
   - Example: ❌ "Find the capital and population of France" → ✅ Split into two sub-questions

2. Retriever Selection Guidelines:
   - `Text Retriever`:
      - For non-visual commonsense knowledge (e.g., "Define photosynthesis")
   - `Text Image Retriever`:
      - When sub-question explicitly references visual elements (e.g., "Describe the painting style of...")
   - `Table Retriever`:
      - For numerical/statistical queries (e.g., "GDP of Japan in 2020")

3. Strict Prohibitions:
   - Never combine multiple questions in <sub-question>
   - Never mention retrieved content in <think>
   - Never select retrievers for non-atomic questions

Examples:
Example 1: Origin question: Which magazine was started first Arthur's Magazine or First for Women?
Assistant: <think>The user wants to compare the starting dates of "Arthur's Magazine" and "First for Women" to see which one
was established first.\n We'll begin by finding out when "Arthur's Magazine" started.\n This information might appear in
textual descriptions.</think>\n<sub-question>When did the magazine 'Arthur's Magazine' start?</sub-question>\n<ret>Text
Retriever</ret>

Example 2: Which magazine was started first Arthur's Magazine or First for Women?\nSub-question1: When did the magazine
'Arthur's Magazine' start?\nAnswer1: 'Arthur's Magazine' was started in 2002
Assistant: <think>We already found that "Arthur's Magazine" was started in 2002. To compare which magazine was established
first, we now need the start date of "First for Women.\n This information might appear in textual descriptions</think>\n<sub-
question>When did the magazine 'First for Women' start?</sub-question>\n<ret>Text Retriever</ret>

Example 3: Origin question: Which magazine was started first Arthur's Magazine or First for Women?\nSub-question1: When did the
magazine 'Arthur's Magazine' start?\nAnswer1:'Arthur's Magazine' was started in 2002\nSub-question2: When did the magazine
'First for Women' start?\nAnswer2: 'First for Women' was started in 1880
Assistant: Sub-question: <think>Now we have both start dates: "Arthur's Magazine" (2002) and "First for Women" (1880). \n
Clearly, "First for Women" was started earlier, so we have enough information to answer the origin question. </think>\n<sub-
question>None</sub-question>\n<ret>None</ret>

Example 4: What is the population of the city that Manchester is in the southeast corner of?
Assistant: <think>The question asks for the population of the city where Manchester is in its southeast corner.\n To identify
that city, we need to check which city is to the northwest of Manchester.\n Names and location details are often found in
textual descriptions, so let's first try the Text Retriever.</think>\n<sub-question>which city is in the northwest of
Manchester</sub-question>\n<ret>Text Retriever</ret>

Example 5: Origin question: What is the population of the city that Manchester is in the southeast corner of?\nSub-question1:
which city is in the northwest of Manchester?\nAnswer1:It's Bolton
Assistant: <think>We found out that the city in question is Bolton.\nTo answer the origin question about population, we now
need the population of Bolton.\n Population data might be available in textual descriptions.</think>\n<sub-question>What is the
population of Bolton?</sub-question>\n<ret>Text Retriever</ret>

Example 6: Origin question: What is the population of the city that Manchester is in the southeast corner of?\nSub-question1:
which city is in the northwest of Manchester?\nAnswer1:It's Bolton\nSub-question2: What is the population of Bolton?\nAnswer2:
It's about 2960K
Assistant: <think>We have identified the city (Bolton) and found its population is about 2960K. That fully answers the origin
question about the population of the city in the southeast corner of Manchester. No further retrieval is needed.</think>\n<sub-
question>None</sub-question>\n<ret>None</ret>

Example 7: Origin question: Who directed the film where Joe Pesci voiced the character of Harry Lyme?
Assistant: <think>The user wants to know who directed the film in which Joe Pesci voiced Harry Lyme. First, we need to identify
the film itself. Information about Joe Pesci's roles should be in textual references.</think>\n<sub-question>What is the name
of the film where Joe Pesci voiced the character of Harry Lyme?</sub-question>\n<ret>Text Retriever</ret>

Example 8: Origin question: When was the saxophonist who recorded A Love Supreme born?
Assistant: <think>We want the birthdate of the saxophonist who recorded "A Love Supreme."\n First, we need to identify the
saxophonist. Because this is general knowledge about music history, a text-based reference should suffice.</think>\n<sub-
question>Who is the saxophonist who recorded A Love Supreme?</sub-question>\n<ret>Text Retriever</ret>

Origin Question: {question}
Reasoning Trajectories : {current_reasoning_trajectories}
```

Figure 6: The Prompt Template of R1-Router (SFT) and R1-Router (Prompt) for Query Generation and Retriever Selection in TextQA tasks. We also use these prompt templates for data construction.

You are a professional question decomposition expert for multi-hop QA systems. Your task is to decompose complex questions into **strictly single-hop** sub-questions and select appropriate retrievers.

**Strict Output Format:**
<think>[Analyze the original question and determine the next required sub-question. Do NOT reveal answers or perform multi-hop reasoning.]</think><sub-question>[Exactly ONE single-hop question one time. If no further information is needed to answer the origin question, write 'None'.]</sub-question><ret> [Choose 1 retriever from: Text Retriever, Text Image Retriever, Table Retriever. Write 'None' if <sub-question> is 'None'.] </ret>

**Critical Rules:**
1.**Atomic Sub-question Definition**:
   - A sub-question is "atomic" only if:
     a) It cannot be further decomposed into simpler questions
     b) It requires exactly **one retrieval action** to answer
     c) Does NOT depend on answers to previous sub-questions
     d) It can be helpful to answer the origin question
   - Example: ❌ "Find the capital and population of France" → ✅ Split into two sub-questions

2. **Retriever Selection Guidelines**:
   - `Text Retriever`:
    - For non-visual commonsense knowledge (e.g., "Define photosynthesis")
   - `Text Image Retriever`:
    - When sub-question explicitly references visual elements (e.g., "Describe the painting style of...")
   - `Table Retriever`:
    - For numerical/statistical queries (e.g., "GDP of Japan in 2020")

3. **Strict Prohibitions**:
   - Never combine multiple questions in <sub-question>
   - Never mention retrieved content in <think>
   - Never select retrievers for non-atomic questions

**Examples:**
**Example 1**: Origin question: which year did he win the world cup champion?
Assistant: <think>We are asked about the year someone won the World Cup. The image might provide a clue, but first we need to identify who this person is. Let's begin by determining the identity of the person in the image.</think></think>\n<sub-question>Who is this person in the image?</sub-question>\n<ret>Text Image Retriever</ret>

**Example 2**: Origin question: which year did he win the world cup champion?\nSub-question1: Who is this person in the image?\nAnswer1: He is Leo Messi.
Assistant: <think>We have identified the person in the image as Lionel Messi. Since the original question asks about the year he won the World Cup, this information may be available both in textual descriptions. We will use the Text Retriever to gather this information.</think>\n<sub-question>Which year did Lionel Messi win the World Cup champion?</sub-question>\n<ret>Text Retriever</ret>

**Example 3**: Origin question: which year did he win the world cup champion?\nSub-question1: Who is this person in the image?\nAnswer1: He is Leo Messi.\nSub-question2: Which year did Lionel Messi win the World Cup champion?\nAnswer2: He won the world cup champion in 2022.
Assistant: <think>We have already identified that Lionel Messi won the World Cup in 2022, which answers the original question. Therefore, no further retrieval or sub-questions are needed.</think>\n<sub-question>None</sub-question>\n<ret>None</ret>

**Example 4**: Origin question: Who is the father of the actor who played the male protagonist in this movie?
Assistant: <think>We are asked about the father of the actor who played the male protagonist in this movie. However, before we can determine the actor's father, we first need to identify which movie this is. The image is likely to provide clues, so we will use the Text Image Retriever to gather both image and text data to identify the movie. This approach allows us to handle both visual and textual information effectively.</think>\n<sub-question>What movie is this poster from?</sub-question>\n<ret>Text Image Retriever</ret>

**Example 5**: Origin question: Who is the father of the actor who played the male protagonist in this movie?Sub-question1: What movie is this poster from?\nAnswer1: The movie's name is Titanic.
Assistant: <think>Now that we know the movie is Titanic, we can focus on finding out who played the male protagonist. This is a straightforward textual query, and we will use the Text Retriever to retrieve this information from relevant sources.</think>\n<sub-question>Who played the male protagonist in Titanic?</sub-question>\n<ret>Text Retriever</ret>

**Example 6**: Origin question: Who is the father of the actor who played the male protagonist in this movie?Sub-question1: What movie is this poster from?\nAnswer1: The movie's name is Titanic.\nSub-question2: Who played the male protagonist in Titanic?\nAnswer2: He is Leonardo DiCaprio.
Assistant:<think>We have already identified the movie as Titanic and the actor who played the male protagonist as Leonardo DiCaprio. The next step is to find out who his father is. Since this is factual information, we will now use the Text Retriever to gather information about Leonardo DiCaprio's father.</think>\n<sub-question>Who is the father of Leonardo DiCaprio?</sub-question>\n<ret>Text Retriever</ret>

**Example 7**: Origin question: Who is the father of the actor who played the male protagonist in this movie?Sub-question1: What movie is this poster from?\nAnswer1: The movie's name is Titanic.\nSub-question2: Who played the male protagonist in Titanic?\nAnswer2: He is Leonardo DiCaprio.\nSub-question3: Who is the father of Leonardo DiCaprio?\nAnswer3: He is George.
Assistant: <think>We have already identified the movie as Titanic, the actor as Leonardo DiCaprio, and his father as George. Therefore, the original question about who is the father of the actor has already been answered. No further retrieval or sub-questions are needed.</think>\n<sub-question>None</sub-question>\n<ret>None</ret>

**Example 8**: Origin question: where is this man's mentor work in?
Assistant: <think>The user is asking about the mentor of the man in the image and where his mentor works. The first step is to identify the man in the image. Once we know who he is, we can proceed to gather information about his mentor. Let's begin by identifying who this man is.</think>\n<sub-question>Who is the person in this image?</sub-question>\n<ret>Text Image Retriever</ret>

Origin Question: {question}
Reasoning Trajectories : {current_reasoning_trajectories}

Figure 7: The Prompt Template of R1-Router (SFT) and R1-Router (Prompt) for Query Generation and Retriever Selection in VQA tasks. We also use this prompt template for data construction.

You are a professional question decomposition expert for multi-hop QA systems. Your task is to decompose complex questions into **strictly single-hop** sub-questions and select appropriate retrievers.

**Strict Output Format:**
<think>[Analyze the original question and determine the next required sub-question. Do NOT reveal answers or perform multi-hop reasoning.]</think><sub-question>[Exactly ONE single-hop question one time. If no further information is needed to answer the origin question, write 'None'.]</sub-question><ret> [Choose 1 retriever from: Text Retriever, Text Image Retriever, Table Retriever. Write 'None' if <sub-question> is 'None'.] </ret>

**Critical Rules:**
1.**Atomic Sub-question Definition**:
   - A sub-question is "atomic" only if:
     a) It cannot be further decomposed into simpler questions
     b) It requires exactly **one retrieval action** to answer
     c) Does NOT depend on answers to previous sub-questions
     d) It can be helpful to answer the origin question
   - Example: ❌ "Find the capital and population of France" → ✅ Split into two sub-questions

2. **Retriever Selection Guidelines**:
   - `Text Retriever`:
     - For non-visual commonsense knowledge (e.g., "Define photosynthesis")
   - `Text Image Retriever`:
     - When sub-question explicitly references visual elements (e.g., "Describe the painting style of...")
   - `Table Retriever`:
     - For numerical/statistical queries (e.g., "GDP of Japan in 2020")

3. **Strict Prohibitions**:
   - Never combine multiple questions in <sub-question>
   - Never mention retrieved content in <think>
   - Never select retrievers for non-atomic questions

**Examples:**
**Example 1**: Origin question: What is the CFL team that has held a position as ol since the 2004 CFL Draft Round 5?
Assistant: <think>The question asks about the CFL team and a position held by an OL since the 2004 CFL Draft. This is very likely to be in a table format, such as a record of team drafts and player positions. We will use the Table Retriever to retrieve relevant structured data from sources like the CFL draft tables.</think>\n<sub-question>What CFL team has held a position as OL since the 2004 CFL Draft Round 5?</sub-question>\n<ret>Table Retriever</ret>

**Example 2**: Origin question: What is the party with a 51.0% Brooklyn vote in the 1932 New York City mayoral election?
Assistant: <think>This question is asking for a specific political party with a 51% vote share in Brooklyn during the 1932 New York City mayoral election. This type of data is likely available in historical election result tables. We will use Table Retriever to retrieve relevant election data from a structured source.</think>\n<sub-question>Which party received 51.0% of the Brooklyn vote in the 1932 New York City mayoral election?</sub-question>\n<ret>Table Retriever</ret>

**Example 3**: Origin question: How many seasons have the Canterbury Bulldogs (8) won in Australian rugby league premieres since 1998 - National Rugby League?
Assistant: <think>The question asks about the number of seasons the Canterbury Bulldogs won in the Australian rugby league since 1998. This data is most likely available in a table, such as National Rugby League championship results. We will use Table Retriever to retrieve the relevant data from structured sources like NRL tables.</think>\n<sub-question>How many seasons have the Canterbury Bulldogs won the Australian rugby league premieres since 1998?</sub-question>\n<ret>Table Retriever</ret>

**Example 4**: Origin question: Among all the races of the Grands Prix of the 1992 Formula 1 season, what are all the places where the date is April 5th?
Assistant: <think>The question asks about specific dates during the 1992 Formula 1 season. The details about each race's location and date are typically stored in a table. We will use Table Retriever to look up the relevant data for the 1992 Formula 1 season and find all the races that took place on April 5th.</think>\n<sub-question>What are all the places where the races of the 1992 Formula 1 season took place on April 5th?</sub-question>\n<ret>Table Retriever</ret>

**Example 5**: Origin question: When was Erskine College founded for the current members of the Conference Carolinas?
Assistant: <think>This question asks about the founding year of Erskine College in relation to its membership in the Conference Carolinas. This is historical data that can be found in  structured records (e.g., college data tables). We will use both Table Retriever to retrieve this information from appropriate sources.</think>\n<sub-question>When was Erskine College founded for the current members of the Conference Carolinas?</sub-question>\n<ret>Table Retriever</ret>

**Example 6**: Origin question: What are all the service dates recorded on 21 November 1945 for the Stena Line Holland BV vessels operated by the Zeeland Steamship Company (SMZ)?
Assistant:<think>This question is asking for service dates for specific vessels on a historical date. These kinds of records are typically available in structured historical tables or logs. We will use Table Retriever to get structured data.</think>\n<sub-question>What are all the service dates recorded on 21 November 1945 for the Stena Line Holland BV vessels operated by the Zeeland Steamship Company?</sub-question>\n<ret>Table Retriever</ret>

**Example 7**: Origin question: What is the total number of players for the Houston Rockets All-Time W Roster, where the years for Rockets are 2004-06?
Assistant: <think>This question involves the Houston Rockets All-Time W Roster from 2004-06, which is most likely recorded in structured sports data tables, such as player rosters. We will use the Table Retriever to retrieve this information from sports databases.</think>\n<sub-question>What is the total number of players for the Houston Rockets All-Time W Roster, where the years for Rockets are 2004-06?</sub-question>\n<ret>Table Retriever</ret>

**Example 8**: Origin question: What is the district for 1952 United States House of Representatives elections, 1988, Massachusetts?
Assistant: <think>This question is asking about a specific district for the 1952 United States House of Representatives elections in Massachusetts in 1988. Election data is often found in both structured tables (election results) and textual descriptions. We will use both Table Retriever to retrieve the election results from structured data.</think>\n<sub-question>What is the district for the 1952 United States House of Representatives elections, 1988, Massachusetts?</sub-question>\n<ret>Table Retriever</ret>

Origin Question: {question}
Reasoning Trajectories : {current_reasoning_trajectories}

Figure 8: The Prompt Template of R1-Router (SFT) and R1-Router (Prompt) for Query Generation and Retriever Selection in TableQA task. We also use this prompt template for data construction.

```
You are a professional question decomposition expert for multi-hop QA systems. Your task is to decompose complex questions into
strictly single-hop sub-questions and select appropriate retrievers.

Strict Output Format:
<think>[Analyze the original question and determine the next required sub-question. Do NOT reveal answers or perform multi-hop
reasoning.]</think><sub-question>[Exactly ONE single-hop question one time. If no further information is needed to answer the
origin question, write 'None'.]</sub-question><ret> [Choose 1 retriever from: Text Retriever, Text Image Retriever, Table
Retriever. Write 'None' if <sub-question> is 'None'.] </ret>

Critical Rules:
1.Atomic Sub-question Definition:
   - A sub-question is "atomic" only if:
     a) It cannot be further decomposed into simpler questions
     b) It requires exactly **one retrieval action** to answer
     c) Does NOT depend on answers to previous sub-questions
     d) It can be helpful to answer the origin question
   - Example: ❌ "Find the capital and population of France" → ✅ Split into two sub-questions

2. Retriever Selection Guidelines:
   - `Text Retriever`:
     - For non-visual commonsense knowledge (e.g., "Define photosynthesis")
   - `Text Image Retriever`:
     - When sub-question explicitly references visual elements (e.g., "Describe the painting style of...")
   - `Table Retriever`:
     - For numerical/statistical queries (e.g., "GDP of Japan in 2020")

3. Strict Prohibitions:
   - Never combine multiple questions in <sub-question>
   - Never mention retrieved content in <think>
   - Never select retrievers for non-atomic questions

Origin Question: {question}
Reasoning Trajectories : {current_reasoning_trajectories}
```

Figure 9: The Prompt Template of R1-Router (Step-GRPO) for Query Generation and Retriever Selection.

```
You are a professional question answering model. Your task is to carefully think through the question based on the information
retrieved and then provide the final answer.

Strict Output Format:
<think>
[Analyze the original question and the retrieved information. Break down the reasoning process step by step. Do NOT provide the
final answer yet.]
</think>
<answer>
[Provide the final answer based solely on the retrieved information.]
</answer>

According to the related information searched, `ter` means this info is from text retriever, `tir` means this info is from text
image retriever, `tar` means this info is from table retriever:{document}\n\n Give me the answer(with the format <answer>
</answer>) to the Question: {question}
```

Figure 10: The Prompt Template of R1-Router for Intermediate Answer Generation. We use this prompt template for all R1-Router models, including R1-Router (SFT), R1-Router (Prompt) and R1-Router (Step-GRPO).

```
You are a professional question answering model. Your task is to carefully think through the question based on the sub-
questions and its answers and then provide the final answer.

Strict Output Format:
<think>
[Analyze the original question and sub-questions with its answers. Break down the reasoning process step by step. Do NOT
provide the final answer yet.]
</think>
<answer>
[Provide the final answer based solely on the information before.]
</answer>

Reasoning Trajectories: {current_reasoning_trajectories}
Based on the information above, give me the final answer of the origin question: {question}
```

Figure 11: The Prompt Template of R1-Router for Final Answer Generation. We use this prompt template for all R1-Router models, including R1-Router (SFT), R1-Router (Prompt) and R1-Router (Step-GRPO).

---

**Vanilla Models**

Q:{question}

Q:{question} Image Caption:{caption}

---

**Vanilla RAG Models**

Below are related information searched, which may be helpful for answering the question later on:{document}
Question: {question}

---

Figure 12: Prompt Templates of Vanilla Models and Vanilla RAG Models.

---

**Examples:**
**User:** Retrieved Passages:\n Hypocrite (Spanish: Hipócrita..!) is a 1949 Mexican thriller film directed by Miguel Morayta\n Q: When did the director of film Hypocrite (Film) die?\n A:
**Assistant:** The film Hypocrite was directed by Miguel Morayta.

**User:** Retrieved Passages:\n Miguel Morayta (15 August 1907 - 19 June 2013) was a Spanish film director and screenwriter.\n\n Q:When did the director of film Hypocrite (Film) die?\n A:The film Hypocrite was directed by Miguel Morayta.
**Assistant:** Miguel Morayta died on 19 June 2013.

**User:** Retrieved Passages:\n Hypocrite (Spanish: Hipócrita..!) is a 1949 Mexican thriller film directed by Miguel Morayta\n Retrieved Passages:\n Miguel Morayta (15 August 1907 - 19 June 2013) was a Spanish film director and screenwriter.\n\n Q: When did the director of film Hypocrite (Film) die?\n A: The film Hypocrite was directed by Miguel Morayta. Miguel Morayta died on 19 June 2013.
**Assistant:** So the answer is: 19 June 2013.

**User:** Retrieved Passages:\n Coolie No. 1 is a 1995 Indian Hindi- language comedy film directed by David Dhawan\n Q: Do director of film Coolie No. 1 (1995 Film) and director of film The Sensational Trial have the same nationality?\n A:
**Assistant:** Coolie No. 1 (1995 film) was directed by David Dhawan.

**User:** Retrieved Passages:\n Coolie No. 1 is a 1995 Indian Hindi- language comedy film directed by David Dhawan\n Retrieved Passages:\n The Sensational Trial is a 1923 German silent film directed by Karl Freund\n Q: Do director of film Coolie No. 1 (1995 Film) and director of film The Sensational Trial have the same nationality?\n A: Coolie No. 1 (1995 film) was directed by David Dhawan.
**Assistant:** The Sensational Trial was directed by Karl Freund.

**User:** Retrieved Passages:\n Coolie No. 1 is a 1995 Indian Hindi- language comedy film directed by David Dhawan\n Retrieved Passages:\n The Sensational Trial is a 1923 German silent film directed by Karl Freund\n Retrieved Passages:\n Karl W. Freund, A.S.C. (January 16, 1890 – May 3, 1969) was a German cinematographer and film director\n Q: Do director of film Coolie No. 1 (1995 Film) and director of film The Sensational Trial have the same nationality?\n A: Coolie No. 1 (1995 film) was directed by David Dhawan. The Sensational Trial was directed by Karl Freund.
**Assistant:** Karl Freund's nationality is Germany.

**User:** Retrieved Passages:\n Coolie No. 1 is a 1995 Indian Hindi- language comedy film directed by David Dhawan\n Retrieved Passages:\n The Sensational Trial is a 1923 German silent film directed by Karl Freund\n Retrieved Passages:\n Karl W. Freund, A.S.C. (January 16, 1890 – May 3, 1969) was a German cinematographer and film director\n Q: Do director of film Coolie No. 1 (1995 Film) and director of film The Sensational Trial have the same nationality?\n A: Coolie No. 1 (1995 film) was directed by David Dhawan. The Sensational Trial was directed by Karl Freund. Karl Freund's nationality is Germany.
**Assistant:** Thus, they do not have the same nationality.

**User:** Retrieved Passages:\n Coolie No. 1 is a 1995 Indian Hindi- language comedy film directed by David Dhawan\n Retrieved Passages:\n The Sensational Trial is a 1923 German silent film directed by Karl Freund\n Retrieved Passages:\n Karl W. Freund, A.S.C. (January 16, 1890 – May 3, 1969) was a German cinematographer and film director\n Q: Do director of film Coolie No. 1 (1995 Film) and director of film The Sensational Trial have the same nationality?\n A: Coolie No. 1 (1995 film) was directed by David Dhawan. The Sensational Trial was directed by Karl Freund. Karl Freund's nationality is Germany. Thus, they do not have the same nationality.
**Assistant:** So the answer is: no.

Retrieved Passages:{document}
Q: {question}
A: {CoT}

---

Figure 13: The Prompt Template of IRCoT Method.

```
You are an intelligent assistant designed to select correct retriever for question.

Instructions:
**Retriever Selection Guidelines**:
   - `Text Retriever`:
     - For non-visual commonsense knowledge (e.g., "Define photosynthesis")
   - `Text Image Retriever`:
     - When sub-question explicitly references visual elements (e.g., "Describe the painting style of...")
   - `Table Retriever`:
     - For numerical/statistical queries (e.g., "GDP of Japan in 2020")
You can only choose 1 retriever from Text Retriever, Text Image Retriever, Table Retriever, or choose none for answering
directly

Current Question: {question}, information before:{document}
```

Figure 14: The Prompt Template of CogPlanner for Retriever Selection.

```
You are an intelligent assistant designed to generate precise sub-questions for a QA system.

Instructions:
1. Context Understanding:
• Question: Analyze the main question provided.
• Image Content(if available): Consider all main entities, objects, and relevant details present in the accompanying image.

2. Sub-question Generation:
• Specificity: Generate one simple and specific sub-question that directly references the relevant entities or objects from
  the image.
• Clarity: Avoid using pronouns or vague terms. Replace all references to entities or objects with their explicit names as
  depicted in the image.
• Relevance: Ensure the sub-question is directly related to retrieving information that will help in answering the main
  question.

3. Output Requirements:
• If a sub-question is necessary for retrieval, output it in the following format:
       ```
       Sub-question: [Your generated sub-question here]
       ```
• If no further retrieval is needed to answer the main question, simply output:
       ```
       Sub-question: None
       ```

Examples:
Example 1: Origin question: Which magazine was started first Arthur's Magazine or First for Women?
Assistant: Sub-question: When did the magazine 'Arthur's Magazine' start?

Example 2: Which magazine was started first Arthur's Magazine or First for Women?\nSub-question1: When did the magazine
'Arthur's Magazine' start?\nAnswer1: 'Arthur's Magazine' was started in 2002
Assistant: Sub-question: When did the magazine 'First for Women' start?

Example 3: Origin question: Which magazine was started first Arthur's Magazine or First for Women?\nSub-question1: When did the
magazine 'Arthur's Magazine' start?\nAnswer1:'Arthur's Magazine' was started in 2002\nSub-question2: When did the magazine
'First for Women' start?\nAnswer2: 'First for Women' was started in 1880
Assistant: Sub-question: None

Example 4: What is the population of the city that Manchester is in the southeast corner of?
Assistant: Sub-question: which city is in the northwest of Manchester

Example 5: Origin question: What is the population of the city that Manchester is in the southeast corner of?\nSub-question1:
which city is in the northwest of Manchester?\nAnswer1:It's Bolton
Assistant: Sub-question: What is the population of Bolton?

Example 6: Origin question: What is the population of the city that Manchester is in the southeast corner of?\nSub-question1:
which city is in the northwest of Manchester?\nAnswer1:It's Bolton\nSub-question2: What is the population of Bolton?\nAnswer2:
It's about 2960K
Assistant: Sub-question: None

Example 7: Origin question: Who directed the film where Joe Pesci voiced the character of Harry Lyme?
Assistant: Sub-question: What is the name of the film where Joe Pesci voiced the character of Harry Lyme?

Example 8: Origin question: When was the saxophonist who recorded A Love Supreme born?
Assistant: Sub-question: Who is the saxophonist who recorded A Love Supreme?

Origin Question: {question}
Sub-question, Answer: {question 1, answer 1, question 2, answer 2,.....question n, answer n}
```

Figure 15: The Prompt Template of IterRetGen for Query Decomposition.

```
Below are related information searched, which may be helpful for answering the question later on:{document}\n\n Give me
the direct answer to the Question: {question}
```

Figure 16: The Prompt Template of IterRetGen for Intermediate Answer Generation.

```
Origin question: {question}
Sub-question, Answer: {question 1, answer 1, question 2, answer 2,.....question n, answer n}

Now, give me the final answer to the origin question: {question}
```

Figure 17: The Prompt Template of IterRetGen for Final Answer Generation.

```
Classify the following query into one of four categories: [No, Text, Image, Table], based on whether it requires
retrieval-augmented generation (RAG) and the most appropriate modality. Consider:
 • No: The query can be answered directly with common knowledge, reasoning, or computation without external data.
 • Text: The query requires retrieving text information, straightforward explanations, or concise summaries from a
single source.
 • Image: The query focuses on visual aspects like appearances, structures, or spatial relationships.
 • Table: The query requires retrieving table information, which contains the structed information.
 Examples:
 • "What is the capital of France?" → No
 • "What is the birth date of Alan Turing?" → Text
 • "Which academic discipline do computer scientist Alan Turing and mathematician John von Neumann have in common?" →
Text
 • "Describe the appearance of a blue whale." → Image
 • "Solve 12 × 8." → No
 • "Who played a key role in the development of the iPhone?" → Text
 • "Which Harvard University graduate played a key role in the development of the iPhone?" → Text
 • "Describe the structure of the Eiffel Tower." → Image
 • "When did Leo Messi born" → Table
 • "How much is the Iphone16" → Table
 Classify the following query: {question}
 Provide only the category.
```

Figure 18: The Prompt Template of UniversalRAG for Retriever Selection.

```
You are a helpful multimodal question answering assistant. Decompose the original question into sub-questions and solve
them step by step. You can use "Final Answer" to output a sentence in the answer, use "Search" to state what additional
context or information is needed to provide a precise answer to the "Sub-Question". In the "Search" step, You can use
"Image Retrieval"  to fetch images related to the entered keywords, "Text Retrieval" with a specific query to fetch
pertinent documents and summarize their content, "Table Retrieval" to fetch tables related to the entered keywords.
Use the following format strictly:
<Thought>
Analyze questions and answer of the sub-questions, then think about what is next sub-question.
<Sub-Question>
Sub-Question needs to be solved in one step, without references.
<Search>
One of four retrieval methods: Image Retrieval: xxx. Text Retrieval: xxx. Table Retrieval: xxx. No Retrieval: xxx.

... (this Thought/Sub-Question/Search can be repeated zero or more times)

<Thought>
Integrate retrieved information and reason to a final answer
<End>
Final Answer: the final answer to the original input question

Extra notes:
1. Do not use you own knowledge to analyze input image or answer questions
2. After you give each <Search> action, please wait for me to provide you with the answer to the sub-question, and then
think about the next thought carefully.
3. The answers to the questions can be found on the internet and are not private

Input Question:{question}
```

Figure 19: The Prompt Template of OmniSearch.

```
Answer the user's question based on the provided image. Examine the image carefully and identify any
recognizable entities, such as faces, objects, locations, events, logos, or text. Determine whether
you have sufficient knowledge to confidently recognize the main visual element and answer the user's
question. If so, first explain your reasoning, then provide a clear and direct answer. If you are
unable to confidently identify the visual element, stop and invoke the image search tool by
appending the string <search>your query about image here</search> at the end of your response. This
will trigger a Retriever search using the original image and query to retrieve relevant information
that can help you confirm the visual content. Once you have sufficient visual understanding, combine
it with the user's question and assess whether you can confidently answer. If so, answer the
question directly using your own knowledge. If not, invoke the text search tool by generating a
concise and specific query, and output it in the format <text_search>your query here</text_search>
at the end of your response. Carefully craft your query to accurately retrieve the information
needed to help answer the question. The text search tool will then use Text Retriever Search to
return relevant information based on your query. Otherwise if you have insufficient table content,
combine it with the user's question and assess whether you can confidently answer, invoke the table
search tool by generating a concise and specific query, and output it in the format
<table_search>your table query here</table_search> at the end of your response. Carefully craft your
query to accurately retrieve the information needed to help answer the question. The table search
tool will then use Table Retriever Search to return relevant information based on your query. You
must include your reasoning inside <reason>...</reason> before taking any action, whether it is
calling the image search tool, generating a text search query, or providing a final answer. The
reasoning may involve analysis of the original image and question, interpretation of search results,
or logical steps leading to the final answer. All search results will be placed inside <information>
and </information> and returned to you. When you are ready to answer the question, wrap your final
answer between <answer> and </answer>, without detailed illustrations.

For example: <answer>Titanic</answer>. Here is the question:{question}
```

Figure 20: The Original First Round Prompt Template of MMSearch-R1.

```
Original question: {question}

Please analyze the search results and the user's question and continue reasoning inside
<reason> and </reason>.
If you are unable to confidently identify the answer, try to use text image search, text search or
table search
If a text search is needed, output the string <text_search>your query here</text_search>
at the end of your response. Please generate a well-crafted query that will help retrieve the most
relevant information.
If a text image search is needed, output the string <image_search>your query here</image_search>
at the end of your response. Please generate a well-crafted query that will help retrieve the most
relevant information.
If a table search is needed, output the string <table_search>your query here</table_search>
at the end of your response. Please generate a well-crafted query that will help retrieve the most
relevant information.
Carefully craft your query to accurately retrieve the information needed to help answer the question.
You must include your reasoning inside <reason>...</reason> before taking any action,
whether it is calling the image search tool, generating a text search query, or providing
a final answer. The reasoning may involve analysis of the original image and question,
interpretation of search results, or logical steps leading to the final answer.
All search results will be placed inside <information> and </information> and returned to you. When
you are ready to answer the question, wrap your final answer
between <answer> and </answer>, without detailed illustrations.

For example: <answer>Titanic</answer>.
```

Figure 21: The Original Intermediate Round Prompt Template of MMSearch-R1.

```
Original question: {question}
Please analyze the search results and the user's question and continue reasoning inside
<reason> and </reason>.
If you determine that additional knowledge is still required to answer the user's question,
stop responding to the question and instead report a warning by outputting the string
"Unable to answer due to lack of relevant information" at the end of your response.
If no further external information is needed, you should provide the final answer by
placing it within <answer> and </answer>. The answer must be concise, clear, and to
the point, without any additional explanation or elaboration.
```

Figure 22: The Final Round Prompt Template of MMSearch-R1.

```
Answer the user's question.

You must always begin with exactly one <reason>...</reason> block BEFORE taking any action.

Requirements for <reason>:
- Only include your reasoning, no actions.
- Briefly justify whether retrieval is needed.
- If retrieval is needed, do ALL of the following INSIDE <reason>:
- Choose EXACTLY ONE retriever from (TEXT, IMAGE, TABLE) and justify it, e.g.: Retriever: TEXT,
because ...
* TEXT → narrative facts, definitions, web/news/PDF/prose
* IMAGE → visual identification (photos, scenes, logos, charts as images)
* TABLE → structured numeric info (stats, prices, schedules, specs)
- Do NOT include any <text_search>/<image_search>/<table_search>, <information>, or <answer> inside
<reason>.

Decision rule AFTER </reason>:
1) If you can confidently answer from current knowledge:
- Output only <answer>...</answer>. Do NOT output any search tag.

2) If you cannot confidently answer:
- Output EXACTLY ONE search tag at the VERY END of your response (last line only), using a precise,
complete query:
* TEXT → <text_search>your query here</text_search>
* IMAGE → <image_search>your query here</image_search>
* TABLE → <table_search>your query here</table_search>
- Write ONE best sub-question in the search tag
- Do NOT output <answer> in the same turn as any search tag.

When search results are later provided inside <information>...</information>:
- Start a NEW <reason>...</reason> that explains which snippets resolved the sub-question and how.
- Then output the final answer ONLY inside <answer>...</answer> (no extra commentary).

Here is the question: {question}
```

Figure 23: The Redisgned First Round Prompt Template of MMSearch-R1.

```
Original question: {question}

You have received search results inside <information>...</information>.
Use ONLY that evidence unless you explicitly state why it is insufficient.

Begin with exactly one <reason>...</reason> block (reasoning only; no actions and NO sub-question
text).
Inside <reason>:
- Summarize what the evidence in <information> implies.
- Resolve conflicts (prefer newer/primary sources) and do any needed calculations/checks.
- Decide "Sufficient to answer" or "Insufficient; need more retrieval".
- If insufficient, CHOOSE EXACTLY ONE retriever TEXT | IMAGE | TABLE and justify the choice.
(Do NOT write the sub-question here. Do NOT output any tags inside <reason>.)

AFTER </reason>:
- If sufficient: output ONLY <answer>...</answer>.
- If insufficient: on the VERY LAST LINE output EXACTLY ONE search tag matching your chosen
retriever, and include the ONE sub-question INSIDE that tag together with the final query, using
this exact inline format (single line, no extra text):
* TEXT → <text_search>your sub-question: ...</text_search>
* IMAGE → <image_search>your sub-question: ...</image_search>
* TABLE → <table_search>your sub-question</table_search>

Guidelines for the tag content:
- The "Sub-question" is the minimal, most answer-unlocking reformulation.
- Do not output <answer> in the same turn as any search tag.
- Output no other text outside the allowed tags.
```

Figure 24: The Redesigned Intermediate Round Prompt Template of MMSearch-R1.

**Question:** How long did Dell Curry play for the Toronto Raptors all-time C roster?
**Ground Truth:** 1999 to 2002

**Exploration Trajectory of R1-Router**
<think> To answer the origin question, we need to determine the duration of Dell Curry's time on the Toronto Raptors all-time C roster. </think>
<sub-question> How long did Dell Curry play for the Toronto Raptors all-time C roster? </sub-question>
<ret> Table Retriever </ret>

Retrieved Documents:
tar-1: [Title] Toronto Raptors all-time roster [Section title] C [Caption] C [Table name] ... [Header] Player [sep]... [sep] Years in Toronto [sep]... [Rows][Row]... [Row] Dell Curry [sep]... [sep] 1999-2002 [sep]...
tar-2:[Title]Toronto Raptors all-time roster[Section title] A [Caption] A [Table name] ... [Header]... [Rows]...
tar-3:[Title]Toronto Raptors all-time roster[Section title] S [Caption] S [Table name] ... [Header]... [Rows]...
<answer> Dell Curry played for the Toronto Raptors all-time C roster for a total of 4 years, from 1999 to 2002. This information is based on the data provided by the table retriever (tar-1), which lists his position as guard and his years in Toronto as 1999-2002, with his school and club team listed as Virginia tech. </answer>
...
<think> The original question sub-questions for the duration of Dell Curry's play for the Toronto raptors all-time c roster. The data provided specifies a total of 4 years, from 1999 to 2002, which is consistent among the sub-questions and answers provided. </think>
<answer> Dell Curry played for the Toronto Raptors all-time C roster for a total of 4 years, from 1999 to 2002.</answer>

Figure 25: A Case Study from R1-Router on Open-WikiTable QA.

**Question:** What city was she born in?
**Ground Truth:** Amherst, Massachusetts

**Exploration Trajectory of R1-Router**
<think> To answer the origin question, we need to identify the person in the image. This requires a visual analysis, which can be achieved using a text image retriever. </think>
<sub-question> Who is the person in the image? </sub-question>
<ret> Text Image Retriever </ret>

Retrieved Documents:
tir-1: Emily Dickinson. Mabel Loomis Todd's daughter, Millicent Todd Bingham, published collec-tions based on the manuscripts held by her mother. tir-2: Emily Dickinson. Daguerreotypetaken at Mount Holyoke, December 1846 or early 1847.
<answer> The person in the image is likely Emily Dickinson, a renowned American poet. </answer>

<think> The next step is to determine the birth city of Emily Dickinson. this requires finding the birthplace of Emily Dickinson. </think>
<sub-question> Where was Emily Dickinson born? </sub-question>
<ret> Text Retriever </ret>

Retrieved Documents:
ter-1: Emily Elizabeth Dickinson was born at the family's homestead in Amherst, Massachusetts, on December 10, 1830. ter-2: Dickinson was born in Amherst, Massachusetts, into a prominent family with strong ties to its community.
<answer> Emily Dickinson was born at the family's homestead in Amherst, Massachusetts, on December 10, 1830. </answer>

<think> The original question has been answered by the sub-question and answer pairs provided. no further information is needed to answer the origin question. </think>
<sub-question> None </sub-question>
<ret> None </ret>

<think> Given the historical context, it is possible to infer that she might be a figure of historical significance, such as Emily Dickinson, who is known for her poetry and lived in Amherst, Massachusetts, during the 19th century. </think>
<answer> The city where she was born is Amherst, Massachusetts. </answer>

Figure 26: A Case Study from R1-Router on Dyn-VQA.

**Question** Where was the place of death of Anastasia Of Serbia's husband?
**Ground Truth:** Hilandar

**Exploration Trajectory of R1-Router**
<think> To answer the origin question, we need to identify the name of Anastasia of Serbia's husband. This requires finding the name of the person she was married to. </think>
<sub-question> What is the name of Anastasia of Serbia's husband? </sub-question>
<ret> Text Retriever </ret>

**Retrieved Documents:**
ter-1: Anastasia of Serbia Anastasia of Serbia Ana (); 1196–d. June 22, 1200) was the princess consort of the Serbian Principality as the wife of Stefan Nemanja (r. 1166–1196). ... ter-2: Anastasia of Serbia was of noble descent. Ana took monastic vows in 1196 and was baptized "Anastasia", after Anastasia of Sirmium. Through the marriage with Stefan Nemanja, they had three sons and three daughters.
<answer> The name of Anastasia of Serbia's husband is Stefan Nemanja. He was the Serbian prince(principality) from 1166 to 1196 and was married to Anastasia of Serbia ... </answer>

<think> The next step is to determine the place of death of Stefan Nemanja, who was the husband of Anastasia of Serbia. </think>
<sub-question> Where did Stefan Nemanja die? </sub-question>
<ret> Text Retriever </ret>

**Retrieved Documents:**
ter-1: Stefan Nemanja monk for some time. They received permission from the new Byzantine emperor to rebuild the abandoned monastery of Hilandar. Death and legacy. Knowing his death was near in his 86th year ... ter-2: Stefan Nemanja stone for his pillow. He died in front of his son Sava and other monks, on 20 February 1199. He was buried in the grounds of Hilandar monastery. His last words requested that Sava take his remains to Serbia ...
<answer> Stefan Nemanja died in front of his son Sava and other monks on 20 February 1199, in the grounds of Hilandar monastery. he was buried in the grounds of Hilandar monastery, which is located in the region of Montenegro .. </answer>
...
<think> The previous answers have all provided substantial information related to Anastasia of Serbia's husband, Stefan Nemanja, including his lifespan, roles, and the place where he passed away. the final answer should be a concise statement reflecting this information. </think>
<answer> The place of death of Anastasia of Serbia's husband, Stefan Nemanja, was in the grounds of Hilandar monastery, located in the region of Montenegro, specifically in the area of Hilandar monastery. </answer>

Figure 27: A Case Study from R1-Router on 2WikiMultihop QA.

