# OpenReview forum: "Query Routing over Multimodal Knowledge Bases for Retrieval-Augmented Reasoning"
_ICLR.cc/2026/Conference — ICLR 2026 Conference Withdrawn Submission_

### Official Review · Reviewer_nSWT · 2025-10-28

**Soundness:** 2
**Presentation:** 3
**Contribution:** 2
**Rating:** 4
**Confidence:** 4

**Summary:**

This paper introduces R1-Router, a framework that performs step-by-step retrieval and routing over multimodal knowledge bases (including text, images, and tables). By applying Step-GRPO, it provides fine-grained rewards at each intermediate reasoning step. The authors claim that the method achieves an average improvement of around 7% over various RAG and routing baselines on multiple QA benchmarks, and further present analyses on routing preferences as well as comparisons of reasoning depth and computational cost.

**Strengths:**

* The paper proposes a new perspective on multimodal RAG, introducing a step-by-step routing and retrieval framework that enhances reasoning over text, images, and tables.

* The training objective is well designed — by integrating step-wise advantage estimation and two types of rewards (query/routing and answer) on top of GRPO, the method effectively mitigates sparse reward issues.

* The empirical evaluation is comprehensive, covering multiple QA scenarios (Text / Visual / Table) and providing clear analyses of routing behavior evolution and computational cost curves, with an average performance gain of around 7% over strong baselines.

**Weaknesses:**

* Novelty: The proposed method builds on an iterative plan–retrieve–reason paradigm and agent-based subproblem decomposition with multimodal routing, which have already been explored in prior work such as OmniSearch[1]. The core difference mainly lies in the step-wise reward and advantage normalization of Step-GRPO, but this appears more like an engineering refinement of GRPO (e.g., grouped normalization, format reward, step-level query/routing reward) rather than a substantial innovation with theoretical support.

* Single evaluation metric: he experiments rely solely on F1-Recall (token-overlap) across all tasks, which does not fully capture the objectives of multimodal, tabular, or extraction-based QA. The lack of comparisons with more appropriate metrics, such as Rouge-L, Retrieval Hit Rate@k, or other task-specific measures.

* Supervision for multiple valid routes: When multiple gold-standard routing paths exist for the same question, it is unclear how r_route handles supervision. Using a single label may be insufficient, and a set-based matching or multi-path reward strategy would be more appropriate.

**Questions:**

Please refer to the weakness section. There are two more questions:

- How do group size,  $\epsilon$, and in Step-GRPO affect performance and stability? Have the authors examined whether reward collapse or variance explosion occurs under different settings?

- How was the upper bound of $n \leq 3$ reasoning steps determined? How does the model perform on datasets that require longer reasoning chains?

---

> ### Author Response · Authors · 2025-11-18
>
> ### **W1: On the novelty of R1-Router**
> Thanks for pointing out the connection to prior work such as OmniSearch. While we share a similar high-level paradigm, there are two main differences. First, systems like OmniSearch and MMSearch-R1 are largely designed for VQA and typically use a **fixed routing pattern** (always search images first, then only use text), which does not transfer well to questions from different modalities. In contrast, R1-Router learns a unified routing policy, deciding at each step whether to retrieve and which KB to query; as shown in Fig. 3, even on VQA it no longer follows a rigid template but adapts its retrieval choices to the specific question. Second, Step-GRPO is designed to jointly optimize the entire long action sequence, alleviating sparse rewards and credit assignment issues that standard single-step GRPO-style methods face in multi-step retrieval–reasoning, rather than being a purely engineering refinement. We will clarify these distinctions more explicitly in the revised version.
>
> ### **W2: On the use of a single evaluation metric.**
> Thank you for the suggestion. Although the tasks span text, multimodal, and tabular settings, they are all formulated as QA with a short textual answer, and we deliberately followed the evaluation protocol of OmniSearch and related work by using F1-Recall (token overlap) as the primary, comparable metric across all datasets. Metrics such as Retrieval Hit Rate@k mainly evaluate the retriever in isolation rather than the end-to-end QA quality, which is the focus of our study.
> To address this concern, we have additionally computed results under another widely used QA metric, CEM, and observe consistent gains. The table below summarizes CEM scores:
> |Method|Open-WikiTables|2WikimultihopQA|InfoSeek|Dyn_VQA|TabFact|WebQA|Avg|
> |-|-|-|-|-|-|-|-|
> |Direct|16.3|41|29.7|19.02|18.1|11.4|22.59|
> |RAG|45.4|41.9|27|3.78|34.9|16.7|28.28|
> |IterRetGen|34.4|45.3|31.8|20.28|38.6|24.7|32.51|
> |IRCoT|35.8|21.2|17.2|12.17|9.6|7.2|17.20|
> |CogPlanner|12.7|43.7|**33.5**|20.1|33.1|26.2|28.22|
> |OmniSearch|5.2|25.4|11.9|6.85|2.3|2.7|9.06|
> |UniversalRAG|27.9|41.7|25.9|3.22|26|17.6|23.72|
> |R1-Router|**50.1**|**50.3**|31|**20.42**|**52.4**|**39.6**|**40.64**|
>
> As shown, R1-Router achieves the best CEM on the most datasets. We will include these CEM results in the revised version.
>
> ### **W3: On multiple valid routing paths and supervision.**
> We agree that, conceptually, there can be multiple valid reasoning and routing paths for the same question. In this work, however, our goal is more pragmatic: to teach the model at least one reliable way to solve each problem. The teacher-generated trajectories provide one concrete, working path from question to answer; learning to follow such a path is already sufficient for the model to acquire the essential reasoning pattern and to reach the correct answer.
>
> ### **Q1: On the effect of group size and ε in Step-GRPO, and whether reward collapse / variance explosion occurs.**
> Thank you for the question. In our implementation, most GRPO-related hyperparameters (including ε for clipping) follow existing GRPO-style settings from prior work, and tuning them is not the focus of this paper. We did observe that a very small group size (e.g., 4) can cause reward collapse in our setting; using a larger group size stabilizes training and improves performance, consistent with observations in BroRL [1] and DAPO [2] that larger groups and slightly higher clipping thresholds can enhance stability. For R1-Router, the configuration already yields good performance without signs of variance explosion under our chosen group size and ε. A more systematic study of these trade-offs is interesting but beyond the main scope of this work;
>
> [1] Hu, et al. BroRL: Scaling Reinforcement Learning via Broadened Exploration
>
> [2] Yu, et al. DAPO: An Open-Source LLM Reinforcement Learning System at Scale
>
> ### **Q2: On the choice of the upper bound $n \le 3$ and longer reasoning chains.**
> The upper bound $n \le 3$ is chosen to match the intrinsic hop count of our benchmarks. Datasets such as Dyn-VQA and 2WikiMultiHopQA are explicitly constructed as multi-hop QA, where the underlying reasoning chains are typically 2–3 hops, so a cap of 3 steps covers the vast majority of cases while preventing degenerate loops. As shown in Fig. 2c, the learned policy often stops before hitting this upper bound once sufficient evidence is gathered.
>
> To the best of our knowledge, there is no widely used multimodal QA benchmark that systematically requires much longer explicit reasoning hops under a comparable setting. We would sincerely appreciate it if you could point out any such datasets; we will include them as baselines, and we are grateful for your help in improving our paper.

---

> > ### Comment · Reviewer_nSWT · 2025-11-26
> >
> > Thanks for the author’s detailed rebuttal.  I will take all additional points into consideration for the final rating. That said, I feel several points need to be further scrutinized:
> >
> > 1. The author mentions that “OmniSearch and MMSearch-R1 are largely designed for VQA and typically use a fixed routing pattern (always search images first, then only use text).” However, in OmniSearch, there is no strict restriction on the retrieval modality, making it a flexible, training-free multimodal agentic RAG baseline. For example, Figure 2(b) in OmniSearch shows image search happening at later stages, and Figure 3 (the pipeline) also does not enforce a fixed modality order. OmniSearch can conduct exactly what you wrote in the rebuttal: “model thinks, generates an intermediate query, selects an appropriate KB or modality, retrieves evidence, and then continues reasoning to produce the final answer.” Therefore, the contribution in this paper appears more like an enhancement of this existing pipeline with a post-training method (basically just using the existing GRPO without substantial changes), rather than a routing training paradigm with some unique designs or analysis in the agentic MM RAG domain. The novelty remains a bit weak to me.
> >
> > 2. The author states that retrieval is not the focus of this paper. However, the way the model decomposes the original question into multiple subqueries can directly influence the retrieval hit rate, which will further affect the overall QA performance. Therefore, I believe adding retrieval-related analyses would further strengthen the paper and make the source of the performance improvement clearer.

---

> ### Author Response · Authors · 2025-11-27
>
> ### **Q1: On the novelty of R1-Router and Step-GRPO.**
> Thank you for the clarification and for pointing out the connection to OmniSearch. We agree that our high-level plan–retrieve–reason paradigm is close in spirit to prior agentic MRAG systems.
>
> Here is our contributions:
> First, Step-GRPO extends GRPO-style methods from single-step or single-turn updates to optimizing the **entire multi-step action sequence**, with decomposed step-wise rewards for intermediate queries, KB/modality routing, and step-level answers, and group-relative normalization over the full trajectory. This directly addresses the sparse reward and credit-assignment issues in multi-hop MRAG, which existing GRPO variants do not explicitly tackle.
>
> Second, R1-Router learns a unified routing and stopping policy over text, text–image, and structured table KBs under large offline corpora. To the best of our knowledge, we are the first to **incorporate a structured table corpus as a first-class KB** in such an agentic MRAG pipeline: the router can explicitly select the table KB when numerical or structured information is needed, which empirically allows the model to answer table- and fact-intensive questions more efficiently than routing only over unstructured text and images.
>
> Third, prior agentic MRAG work, such as OmniSearch is primarily evaluated in the VQA domain, whereas our framework is instantiated and tested on multiple modalities and task types (text QA, table QA, and VQA) under a unified setting. This demonstrates that the proposed routing + Step-GRPO framework generalizes beyond pure VQA, and suggests that it can be further extended to even richer all-modality QA scenarios in future work.
>
> Finally, our experiments show that R1-Router is the only method among all w/ KB routing baselines that can match or even slightly surpass its own w/o KB routing variant (i.e., retrieving from all KBs simultaneously). This indicates that R1-Router can accurately select the appropriate KB for each step of reasoning, whereas other baselines, despite taking KB routing as their core design, consistently underperform their corresponding w/o routing (all-KB) settings. Here we exclude OmniSearch w/o KB routing, since feeding all retrieved information as context causes its input to exceed the maximum model length (8192 tokens), leading to frequent inference failures.
> |Method|Strategy|Open-WikiTables|2WikimultihopQA|InfoSeek|Dyn_VQA|TabFact|WebQA|Avg|
> |-|-|-|-|-|-|-|-|-|
> |CogPlanner|w/ KB routing|16.50|49.28|42.23|36.60|33.10|84.82|43.76|
> |CogPlanner|w/o KB routing|38.95|50.99|40.94|36.08|38.60|84.19|48.29|
> |OmniSearch|w/ KB routing|7.72|31.02|24.45|18.94|2.30|58.02|23.24|
> |UniversalRAG|w/ KB routing|31.12|47.30|37.25|17.91|26.00|79.48|38.84|
> |UniversalRAG|w/o KB routing|49.99|48.31|39.06|14.12|34.90|76.63|43.84|
> R1-Router|w/ KB routing|53.95|**55.47**|**43.60**|**39.24**|**52.40**|**90.92**|**55.93**
> R1-Router|w/o KB routing|**56.65**|50.26|41.20|38.96|51.80|90.90|54.96
>
> ### **Q2: On retrieval-related analysis and the source of performance gains.**
> Thank you for the insightful comment. While our main focus is on the end-to-end QA performance, we agree that the way the model decomposes questions and issues subqueries is closely tied to retrieval quality, and we have already included several retrieval-related analyses. In particular, Fig. 2c shows that R1-Router with Step-GRPO can locate answer-bearing evidence in fewer reasoning steps than other variants, indicating that it learns to formulate effective subqueries and stop earlier once sufficient information is obtained. For completeness, we have also added a more detailed horizontal comparison of the average number of steps:
>
> |Method|TableQA|VisualQA|TextQA|
> |-|-|-|-|
> |CogPlanner|1.02|1.11|1.38|
> |OmniSearch|1.06|1.53|1.28|
> |MMSearch-R1|1.09|1.31|1.49|
> |R1-Router(Prompt)|1.09|1.55|1.45|
> |R1-Router(SFT)|1.11|1.35|1.40|
> |R1-Router(Step-GRPO)|1.01|1.11|1.46|
>
> In addition, the comparison between w/ and w/o KB routing further highlights the quality of our routing and, indirectly, the quality of retrieval: R1-Router with KB routing is the only routing-based method in our study that can match or slightly surpass its own all-KB variant, whereas other baselines see their w/ KB routing versions lag behind w/o routing. This indicates that R1-Router not only improves reasoning, but also learns to choose more appropriate KBs, retrieve less noisy evidence, and thus improves downstream QA.
>
> If you have specific suggestions for more retrieval analyses or diagnostics, we would be very glad to incorporate them and run the corresponding experiments.

---

### Official Review · Reviewer_BqxX · 2025-11-01

**Soundness:** 2
**Presentation:** 3
**Contribution:** 2
**Rating:** 4
**Confidence:** 3

**Summary:**

The paper proposes R1-Router, a framework for Multimodal Retrieval-Augmented Generation (MRAG) that dynamically decides when and where to retrieve external knowledge during reasoning. Unlike prior static or heuristic retrieval pipelines, R1-Router employs a reasoning-driven query routing mechanism, combined with a reinforcement learning algorithm called Stepwise Group Relative Policy Optimization (Step-GRPO). The introduction motivates this need by arguing that current MRAG systems treat retrieval as a one-shot process, neglecting dynamic reasoning and adaptive querying across heterogeneous multimodal knowledge bases (text, image, table).

**Strengths:**

- Propose R1-Router and Step-GRPO, a reasoning-aware MRAG controller that adaptively queries multiple modalities.
- Extensive experiments across six benchmarks demonstrate +7% improvement and robust generalization.

**Weaknesses:**

- From the method perspective, Step-GRPO relies heavily on ground-truth reasoning trajectories that are constructed using large teacher models (specifically R1-Distill-Qwen-32B and Qwen2.5-VL-7B). While this provides a strong supervisory signal during training, it also introduces potential distillation bias. Since the teacher-generated trajectories already embed the reasoning style and retrieval preferences of these large models, the student (R1-Router) may simply imitate the teachers’ patterns rather than discovering genuinely new reasoning behaviors. This dependency limits the generality and autonomy of Step-GRPO.

- In the method design, the Step-GRPO and R1-router introduce lots of hyperparameters, such as different reward designs. A thorough analysis or ablation of these parameters is missing.

- Related work on advanced agentic search is missing.

**Questions:**

Could retrieval stopping be learned instead of fixed n≤3?

Is retrieval redundancy or cost considered in the reward function?

Does Step-GRPO scale to larger KBs or unseen modalities?

---

> ### Author Response · Authors · 2025-11-18
>
> ### **W1: On dependence on ground-truth reasoning trajectories and potential distillation bias.**
> Thank you for raising this point. In our framework, golden trajectories are used as a weak prior and reward shaping signal, not for strict behavior cloning. During Step-GRPO training, the policy at each step still samples its own intermediate query, routing decision, and step answer; the reward combines answer correctness, routing correctness, and a soft alignment term with the teacher trajectory. We do not apply cross-entropy on golden thoughts or force the think content to match—higher similarity gives slightly higher shaping reward, but alternative actions that lead to correct evidence and answers are also rewarded.
>
> Empirically, R1-Router surpasses its own teacher on several benchmarks. For example, in VQA, the teacher used to construct trajectories is R1-Router (prompt) in Table 2, yet the R1-Router (Step-GRPO) attains higher accuracy. This indicates that the our method is not simply copying the golden reasoning style, but can refine and improve upon it. In practice, golden trajectories serve as scaffolding to guide exploration in a large action space, while still leaving room for the policy to discover more effective reasoning behaviors.
>
> ### **W2: On hyperparameters in Step-GRPO and R1-Router.**
> Most hyperparameters follow existing GRPO-style settings （e.g., Visual-RFT[1]）, and all baselines use the same configuration, so tuning them is not the focus of this work. Step-GRPO only introduces a few extra reward coefficients (e.g., $\alpha$,$\beta$ in Eq. (9)). In practice, the router-related reward quickly saturates to near 1.0, so $\alpha$,$\beta$ mainly rescale an already stable signal and have little impact on stability or final performance within a reasonable range.
>
> [1] Liu et al., Visual-RFT: Visual Reinforcement Fine-Tuning.
> ### **W3: On “advanced agentic search” in the multimodal setting.**
> Thank you for the suggestion. In our experiments and related work, we have already compared against recent multimodal MRAG systems with KB routing, including OmniSearch, MMSearch-R1, and other baselines that explicitly perform KB routing. To the best of our knowledge, these are the main multimodal methods that combine agentic-style routing with cross-KB retrieval. We would sincerely appreciate it if you could point out any relevant methods, we will include them as baselines, and we are grateful for your help in improving our paper.
> ### **Q1: On learning retrieval stopping vs. using a fixed upper bound $n \le 3$.**
> We agree that retrieval stopping can and should be learned. In fact, as stated around line 304, the bound $n \le 3$ is only used as a safety cap to prevent the model from entering degenerate loops when no useful information can be found; the stopping decision itself is learned by the policy. Concretely, the model is rewarded for producing a correct final answer with as few unnecessary steps as possible, and it can terminate early once sufficient evidence has been collected.
> This behavior is reflected in our results: Fig. 2c reports the average number of steps at which the model already finds answer-bearing evidence, showing that it typically stops before hitting the upper bound; and in Fig. 26 we provide a qualitative case where the model actively stops retrieving after gathering enough information.
> ### **Q2: On whether retrieval redundancy or cost is considered in the reward function.**
> Thank you for the question. We do not add an explicit cost term in the Step-GRPO objective. Instead, redundancy and cost are controlled implicitly. The policy is encouraged to stop once sufficient evidence is gathered, since extra steps that do not improve the final answer bring no additional reward, and we also impose a small safety cap n \le 3 to avoid loops. As a result, R1-Router uses fewer retrieval steps on average than other methods while achieving higher accuracy (Fig. 2c). In addition, all experiments use three fixed offline corpora with dense retrieval, rather than repeated online web search, which further reduces retrieval redundancy and system-level cost.
> ### **Q3: On whether Step-GRPO scales to larger KBs or unseen modalities.**
> Algorithmically, Step-GRPO is agnostic to KB size and modality: it learns a step-wise policy over which KB to call and how to query it, and does not rely on any particular index scale or feature type. The practical bottleneck lies in the dense retriever, not in the Step-GRPO update. Empirically, our KBs are already large-scale: as shown in Appendix A.2, the text KB is the full English Wikipedia dump (enwiki-20241020, ≈52M passages) and the text–image KB is M-BEIR (over 5.6M candidates), showing that R1-Router with Step-GRPO can handle realistic KB sizes. For unseen modalities, each modality-specific retriever is simply another action in the routing space, so adding a new modality (e.g., audio or video) amounts to plugging in a new retriever and defining its step-wise rewards.

---

### Official Review · Reviewer_pcKT · 2025-11-01

**Soundness:** 2
**Presentation:** 3
**Contribution:** 2
**Rating:** 4
**Confidence:** 2

**Summary:**

This paper proposes R1-Router, a Multimodal Retrieval-Augmented Generation (MRAG) framework enabling Large Multimodal Language Models (MLLMs) to dynamically decide when and where to retrieve knowledge during reasoning. The model introduces Stepwise Group Relative Policy Optimization (Step-GRPO), i.e., a reinforcement-learning variant assigning step-specific rewards to intermediate reasoning stages. Experiments across 6 QA datasets show promising performance.

**Strengths:**

- The motivation is clear. Dynamic query routing over heterogeneous multimodal KBs is necessary to break through the limit of language only.

- Step-GRPO extends GRPO with fine-grained stepwise rewards for multi-stage reasoning.

- Dataset selection is extensive, which covers text, vision, and tables.

- The paper is well-organized with appendix details on retrievers and training configs.

**Weaknesses:**

- Figure 1 is not very self-illustrative; it is not very easy to distinguish the main breaking novelty or uniqueness of the proposed method.

- The theoretical contribution of the paper is not extensive. Step-GRPO is currently empirical; a convergence or variance analysis versus standard GRPO would add credibility and novelty to the paper.

- Training cost is not very clearly shown in the paper, which may be a concern.

- The design of Eq (9) can be extended, especially the relationship between 'ask' and 'route', the need for balancing, whether they are contributing or competing with each other.

**Questions:**

Beyond weakness, there are also some questions below.

- How sensitive is Step-GRPO to $\alpha$ and $\beta$?

- What percentage of the generated reasoning trajectories are filtered out as incorrect during training (Appendix A.3)?

- How would R1-Router perform if KB modalities contain noisy or partially missing entries (e.g., imperfect captions)?

---

> ### Author Response · Authors · 2025-11-18
>
> ### **W1: Regarding Fig. 1 and the uniqueness of our method.**
> The core novelty of our approach is that we unify query decomposition, KB routing,  information retrieval, and final answering into a single LRM-driven iterative loop, where the model thinks, generates an intermediate query, selects an appropriate KB or modality, retrieves evidence, and then continues reasoning to produce the final answer.
> At each step, the model dynamically decides whether to retrieve and which knowledge base to query based on its current reasoning state, instead of using one-shot retrieval or a fixed routing pattern over pre-decomposed sub-queries. This unified decision process of multi-step thinking with learned dynamic KB routing, as illustrated in Fig. 1(c), is the main distinguishing aspect of our method, and we will revise Fig. 1 to better highlight this.
> ### **W2: On the theoretical contribution.**
> Thank you for the comment. Step-GRPO is positioned mainly as an algorithmic extension of the GRPO family for multi-step retrieval-augmented reasoning, rather than a full theoretical framework. Standard GRPO variants (e.g., DAPO, GSPO) still update the policy per single action/step, which in our setting suffers from highly sparse rewards (mostly on the final answer) and high-variance, non-smooth returns across steps, making training unstable and credit assignment to intermediate decisions difficult.
> Step-GRPO instead treats the entire multi-step decision process as a structured action sequence: at each step we introduce decomposed step-wise rewards (for intermediate query, routing, and step answer), and apply group-relative normalization over the full trajectory. This effectively densifies and smooths the reward signal along time, which empirically reduces variance and stabilizes training. As shown in Table 2 and Fig. 2(a), standard GRPO-style baselines exhibit more unstable reward curves and less consistent performance gains, whereas Step-GRPO leads to steadily improving rewards and significantly stronger reasoning performance.
> ### **W3: On training cost.**
> Thank you for raising this concern. Concretely, training our main R1-Router model with Step-GRPO requires about 400 GPU-hours on NVIDIA A100-80GB GPUs. We believe this compute budget is moderate and comparable to existing GRPO-style RL setups for large models, and we will add these details to the implementation section for transparency.
> ### **W4 & Q1: On Eq. (9), the relationship between “ask” and “route”, and the sensitivity to $\alpha$ / $\beta$.**
> Thank you for the question. In Eq. (9), the “ask” and “route” terms are intended as complementary shaping signals on the same step-wise decision, rather than competing objectives: the ask term encourages the model to propose clear, informative follow-up queries, while the route term encourages choosing an appropriate retriever/KB for that query so that the multi-step reasoning remains coherent. Both push the policy toward better use of the action sequence, and we did not observe any conflicting behavior between them in practice.
> Empirically, the router-related reward quickly saturates (it approaches ≈1.0 early in training), so the routing component soon provides relatively stable gradients. Consequently, Step-GRPO is not very sensitive to the exact values of $\alpha$ and $\beta$ within a reasonable range: changing them mainly rescales an already stable signal and does not noticeably affect training stability or final performance.
> ### **Q2: On the percentage of filtered reasoning trajectories (Appendix A.3).**
> Thank you for the question. In our data construction, we apply automatic filters that require the final answer to match the gold label and all intermediate steps to pass our consistency checks. Under these criteria, we filter out roughly 60% of the generated trajectories for the text and table training datasets (generated by R1-Distill-Qwen-32B), and about 75% of the trajectories for the image datasets (generated by Qwen-2.5-VL-7B). We will add these concrete ratios to Appendix A.3 for clarity.
> ### **Q3: On robustness to noisy or partially missing KB entries.**
> Thank you for the question. Our multimodal KBs are indeed noisy: many image entries come with paraphrastic or partially mismatched captions. The case study in Sec. 5.4 directly illustrates how R1-Router can handle this: when the retrieved image–text pairs contain mixed, noisy descriptions, the model does not simply trust all captions. Instead, it first grounds on the original image via the VL backbone, and uses this visual evidence to reverse-check which retrieved description is actually compatible with the image (e.g., identifying the bird in the picture as a “Bay-breasted Warbler” despite surrounding noisy candidates). Only after this visual grounding step does it proceed to the next reasoning step, where it routes to the text KB to retrieve clean taxonomic facts and finalize the answer.

---

### Official Review · Reviewer_gM5A · 2025-11-02

**Soundness:** 3
**Presentation:** 2
**Contribution:** 3
**Rating:** 4
**Confidence:** 3

**Summary:**

I found the core contribution, the Step-GRPO algorithm, to be a significant strength and a clear leap beyond static RAG. One of my issue is the model's dependence on golden trajectories from a teacher, which limits it to imitation rather than true discovery. I also question the claim of improved efficiency, as the accuracy gains appear to come at a substantial cost in inference time. Finally, the reported performance for some of the baseline is drastically different from its original paper, which undermines my confidence in the evaluation.

**Strengths:**

The Step-GRPO algorithm is a major highlight. By designing fine-grained rewards for each decision point in the reasoning process (sub-question generation, KB routing, intermediate answer generation), it effectively addresses the reward sparsity and credit assignment challenges faced by traditional RL in long-sequence, multi-action tasks. The paper is clearly written and well-structured. The method decides not only "what" to retrieve but also "when" and "from where," representing a significant leap beyond the traditional static RAG paradigm.

**Weaknesses:**

1. Dependence on "Golden Reasoning Trajectories": The Training relies on "golden reasoning trajectories" generated by a more powerful teacher model. This means the model's performance ceiling might be constrained by the quality of this synthetic data. The model may be biased towards imitating the teacher's specific reasoning patterns rather than discovering novel, potentially superior, paths.
2. As stated in the end of the abstract “…enhance both efficiency and accuracy”, improved accuracy can be seen in Sec. 5, however, the improved efficiency lacks detailed illustration. Why does “2s additional per-step latency align well with performance-computational-cost-trade-off” (Ling 410)? E.g., compared to CogPlanner, the F1-Recall increase around 25% while the inference time increases around 60%.

**Questions:**

1. As reported in [OmniSearch paper](https://arxiv.org/pdf/2411.02937) Table 4, OmniSearch obtains 41.20 on Dyn-VQA. However, in this paper, OmniSearch obtains 18.94 (Table 1). What causes this significant gap? (Did I miss something?) If KB routing degrades the model, what is the performance of OmniSearch (or other baselines) w/o KB routing?
2. Why does random routing with Step-GRPO obtain comparable performance (Table 2, 52.50 & 55.93)? Could you please elaborate on “random routing”? E.g., we may adopt Text Retrieval (selected by random) even if the router predicts to use Text-Image Retrieval?
3. Figure 2c: how many steps do other vanilla baseline methods use (E.g., CogPlanner and OmniSearch)?

---

> ### Author Response · Authors · 2025-11-18
>
> ### **Response to W1:**
> Thank you for raising this concern about our use of “golden reasoning trajectories.” We would like to clarify:
> Golden trajectories as a weak prior, not behavior cloning. At each step, we include a piecewise shaping reward for the semantic alignment between the model’s follow-up query/router choice and a golden trajectory (higher similarity to higher shaping score). However, this is only one component of the total return. The reward also includes routing correctness and step-level answer correctness, which are directly tied to task performance. We do not constrain the think content or its form; the policy is free to derive its own next-step rationale.
>
> Why this prior is reasonable. Our training data provides a correct answer and at least one reachable correct trajectory. Early in training, this weak prior narrows the search space and accelerates convergence; as training progresses, the policy can build on it to learn more flexible, equivalent paths rather than copying surface forms. Empirically, we outperform SFT and exhibit more adaptive modality/step selection (see Fig. 3), indicating learned, instance-dependent selectivity rather than rote imitation.
> ### **Response to W2:**
> Thank you for the suggestion. By “efficiency”, we hit question-relevant evidence faster and issue fewer retrieval calls. As shown in Fig. 2c, our policy reaches answer-bearing evidence in fewer steps and stops early, while accuracy also improves; therefore, the claim that we enhance both efficiency and accuracy holds.
>
> Regarding the concern that “compared to CogPlanner, F1-Recall increases by ~25% while inference time increases by ~60%”:
> Relative ratios are not an appropriate metric. When the baseline latency is small (e.g., 0.1 s), an absolute increase of 0.1–0.2 s is inflated to “+100%–200%,” which does not faithfully reflect the performance-computational-cost-trade-off. A more informative report should also consider absolute magnitudes: in our measurements, about 1.8 s of marginal overhead yields a 12% improvement. For most RAG scenarios, this “small, controllable delay for a substantial quality gain” is acceptable and practically valuable.
> ### **Response to Q1:**
> The two OmniSearch numbers are obtained under different retrieval regimes, so they are not directly comparable. In the original OmniSearch paper, Dyn-VQA is evaluated with online web-scale retrieval (multi-source, continuously updated). In our unified and fair evaluation, all methods, including OmniSearch, are run on three fixed offline KBs with their own retrievers. Offline corpora are naturally weaker in coverage and freshness than online search, especially for a time-sensitive, long-tail dataset like Dyn-VQA, so the answerability of many questions drops and all methods’ absolute scores decrease accordingly. Therefore, 41.20 vs 18.94 reflects a change in retrieval setting, not a degradation caused by our method.
> Regarding w/o KB routing, under our offline setup the correspondences are:
> UniversalRAG (w/o KB routing) equals to RAG (w/ all KBs), which replaces routing with a single “all-KB” retriever; CogPlanner (w/o KB routing) equals to IterRetGen (w/ all KBs). The corresponding OmniSearch (w/ w/o routing) and R1-Router (w/o routing) experiments are currently running and will be added to the updated version once completed. KB routing does indeed lead to a performance drop for these baselines, and since KB routing is precisely their core design focus, this also highlights certain limitations of existing baseline methods.  We hope this clarifies the setting difference, and are happy to address any further questions in the meantime.
> ### **Response to Q2:**
> Thank you for the question. Random routing means that at each step the model still generates an intermediate thought and a follow-up query, but during retrieval we ignore the model’s predicted retriever choice and instead random from the three retrievers (Text / Text-Image / Table) to search for answers. All other parts of the inference procedure remain the same.
>
> Under the configuration of at most 3 iterative steps, even with random routing, when the model does not obtain sufficient evidence in early steps it will rewrite/refine the query and try again, so the probability of “hitting” a suitable retriever within three random trials is not low.
> ### **Response to Q3:**
> Thank you for the question. In Fig. 2c we only plotted R1-Router variants for clarity, but we did measure the average number of reasoning steps for other baselines as well. Under the same upper bound $n \le 3$, the average steps are:
> |Method|TableQA|VisualQA|TextQA|
> |-|-|-|-|
> |CogPlanner|1.02|1.11|1.38|
> |OmniSearch|1.06|1.53|1.28|
> |MMSearch-R1|1.09|1.31|1.49|
> |R1-Router(Prompt)|1.09|1.55|1.45|
> |R1-Router(SFT)|1.11|1.35|1.40|
> |R1-Router(Step-GRPO)|1.01|1.11|1.46|

---

> > ### Author Response · Authors · 2025-11-23
> >
> > For OmniSearch w/o KB routing, under our offline all-KB setting we need to concatenate the full interaction history with all retrieved passages. In practice this often makes the input exceed the model’s maximum context length of 8192 tokens, leading to decoding failures on a large portion of all benchmarks. This further illustrates that KB routing is practically important for controlling context length and mitigating long-context issues in such setups.
> >
> > We also evaluated R1-Router w/o KB routing, where all queries are sent to three KB retrievers and retrurn 15 retrieved passages. The average F1-Recall across benchmarks is:
> > |Method|Open-WikiTables|2WikimultihopQA|InfoSeek|Dyn_VQA|TabFact|WebQA|Avg|
> > |-|-|-|-|-|-|-|-|
> > w/ KB routing|53.95|**55.47**|**43.60**|**39.24**|**52.40**|**90.92**|**55.93**
> > w/o KB routing|**56.65**|50.26|41.20|38.96|51.80|90.90|54.96
> >
> > As shown, R1-Router with KB routing performs better than its w/o routing, in contrast to previous baselines where w/o KB routing tends to outperform their routed versions. This indicates that, once the model has stronger step-wise reasoning ability, it can learn to select the appropriate KB during reasoning, benefit from fewer and more focused retrieved contents, and thereby reduce noise and improve performance. These results further demonstrate the effectiveness of our R1-Router design.

---

### Comment · Area_Chair_mF9F · 2025-11-22
**Official Comment by Area Chair mF9F**

Dear Authors and Reviewers,

I would like to thank the authors for providing detailed rebuttal messages on time.

To reviewers: I would like to encourage you to carefully read all other reviews and the author responses and engage in an open exchange with the authors. Please post your first response as soon as possible within the discussion time window. Ideally, all reviewers will respond to the authors, so that the authors know their rebuttal has been read.

Best regards,
AC

---

### Comment · Area_Chair_mF9F · 2025-11-25

Dear Reviewers,

The authors have responded to your reviews. Please review and provide your feedback and responses.

Best,

Your AC

---

### Comment · Area_Chair_mF9F · 2025-11-27

Dear Reviewers,

Thank you for your valuable reviews. With the Reviewer-Author Discussions deadline approaching, please take a moment to read the authors' rebuttal and the other reviewers' feedback, and participate in the discussions and respond to the authors. Finally, be sure to complete the "Final Justification" text box and update your "Rating" as needed. Your contribution is greatly appreciated. I will flag irresponsible (final) reviews and/or any reviewers not participating in discussions.

Reviewers are expected to stay engaged in discussions, initiate them and respond to authors’ rebuttal, ask questions and listen to answers to help clarify remaining issues.

It is not OK to stay quiet.

It is not OK to leave discussions till the last moment.

If authors have resolved your (rebuttal) questions, do tell them so.

If authors have not resolved your (rebuttal) questions, do tell them so too.

Thanks.

AC

---

### Note · Authors · 2026-01-22

I have read and agree with the venue's withdrawal policy on behalf of myself and my co-authors.